# Grounding and Enhancing Informativeness and Utility in Dataset Distillation

**Shaobo Wang**[†1,2] **Yantai Yang**[1] **Guo Chen**[1] **Peiru Li**[2] **Kaixin Li**[3] **Yufa Zhou**[4]
**Zhaorun Chen**[5] **Linfeng Zhang**[†1,2]

[1]EPIC Lab, SJTU  [2]Shanghai Jiao Tong University  [3]National University of Singapore
[4]Duke University  [5]The University of Chicago  [†]Corresponding authors
E-mail: {shaobowang1009,zhanglinfeng}@sjtu.edu.cn
Code: https://github.com/gszfwsb/InfoUtil

## ABSTRACT

Dataset Distillation (DD) seeks to create a compact dataset from a large, real-world dataset. While recent methods often rely on heuristic approaches to balance efficiency and quality, the fundamental relationship between original and synthetic data remains underexplored. This paper revisits knowledge distillation-based dataset distillation within a solid theoretical framework. We introduce the concepts of Informativeness and Utility, capturing crucial information within a sample and essential samples in the training set, respectively. Building on these principles, we define *optimal dataset distillation* mathematically. We then present InfoUtil, a framework that balances informativeness and utility in synthesizing the distilled dataset. InfoUtil incorporates two key components: (1) game-theoretic informativeness maximization using Shapley Value attribution to extract key information from samples, and (2) principled utility maximization by selecting globally influential samples based on Gradient Norm. These components ensure that the distilled dataset is both informative and utility-optimized. Experiments demonstrate that our method achieves a 6.1% performance improvement over the previous state-of-the-art approach on ImageNet-1K dataset using ResNet-18.

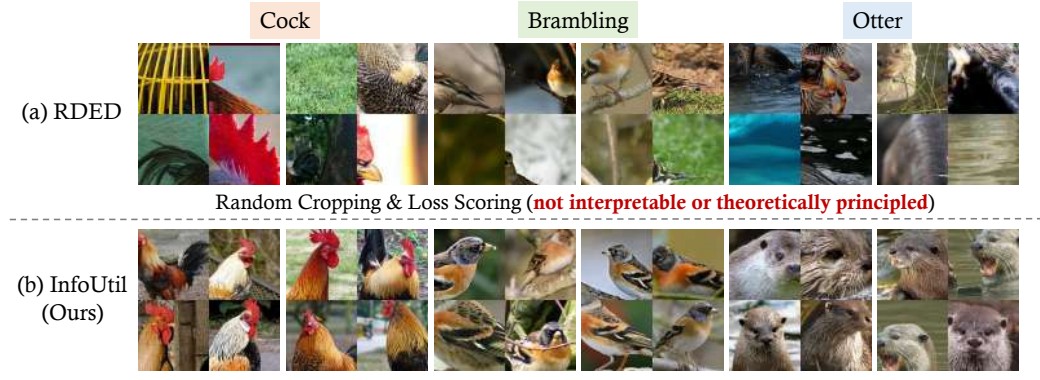

Figure 1: Comparison of visualization results between previous method (a) RDED (Sun et al., 2024) and (b) our InfoUtil. Unlike prior methods relying on random selection and intuitive scoring, InfoUtil is both interpretable and theoretically grounded. It synthesizes images that more accurately capture semantically meaningful regions with principled scores. Prioritizing core content over irrelevant details like background elements ensures a more focused and meaningful representation.

## 1 INTRODUCTION

Dataset distillation (DD) (Wang et al., 2018; Sachdeva & McAuley, 2023) has emerged as a promising approach for enabling vision models to achieve performance comparable to training on large datasets, but with only a small set of synthetic samples. The core idea behind DD is to compress large

Figure 2: InfoUtil's pipeline for *optimal dataset distillation* involves two key steps: (i) Step 1 maximizes informativeness via the Shapley Value (a game-theoretic attribution method), retaining the most informative patches to form compressed samples. (ii) Step 2 maximizes utility by scoring these candidates with a judge model—using Gradient Norm (proven as a utility upper bound)—and retaining top samples. The final distilled dataset contains only the most informative, high-utility compressed samples. Image reconstruction and soft label generation phases are omitted here.

datasets by synthesizing and optimizing a smaller, representative dataset. Models trained on distilled dataset are expected to match the performance of those trained on the original, larger dataset.

Currently, two primary lines of approaches are used to tackle DD: *i.e.*, matching-based methods (Wang et al., 2018; Zhao & Bilen, 2022; Zhao et al., 2021; Cazenavette et al., 2022; Zhou et al., 2022), which aim to align the performance between the distilled dataset and the original dataset by matching gradients, features, distributions, or trajectories, and knowledge distillation-based methods (Yin et al., 2023; Shao et al., 2024), which decouple dataset distillation into two stages. In the first stage, the real data is compressed into a teacher model. In the second stage, the teacher model transfers knowledge to the distilled images through deep inversion-like methods (Yin et al., 2020). Despite their success, these existing methods face two challenges:

> **Challenge 1: Efficiency-Performance Trade-off.** *Most matching-based methods require significant GPU memory and time, making them impractical for real-world applications.*

For bi-level matching-based methods, the key challenge lies in the trade-off between performance and efficiency (Zhao et al., 2021; Zhao & Bilen, 2021; Lee et al., 2022; Wang et al., 2024a; Guo et al., 2023; Cui et al., 2023). For example, the state-of-the-art (SOTA) trajectory matching method (Guo et al., 2023) requires more than 4 NVIDIA A100 80GB GPUs to synthesize a 50 image-per-class (IPC) dataset on Tiny-ImageNet. Such high resource demands severely limit scalability of these methods, making it extremely challenging to apply to larger datasets like ImageNet-1K.

For knowledge distillation-based methods, although they often perform better, the lack of a solid theoretical foundation impairs their interpretability (Yin et al., 2023; Shao et al., 2024; Sun et al., 2024) and prevents a principled solution. This limitation leaves practitioners with limited insight into why certain samples are selected for compression or how the distillation process relates to underlying data. Therefore, despite demonstrating impressive empirical results, they fall short in providing the transparency required for high-stakes or regulated applications.

> **Challenge 2: Lack of Interpretability.** *Current methods are largely heuristic, lacking a principled framework to ensure the resulting distilled datasets are interpretable.*

To rethink previous methods within a principled framework, we reconsider the knowledge distillation-based dataset distillation process by introducing *Optimal Dataset Distillation* (Definition 4). The concept is built on *Informativeness* (Definition 1) and *Utility* (Definition 3) for desired distilled dataset. Intuitively, *Informativeness* captures essential information in each sample, while *Utility* reflects the importance of each sample for model training, whether included or excluded.

Built on the theoretical framework, we propose InfoUtil, **Info**rmativeness and **Util**ity-enhanced Dataset Distillation (InfoUtil), a method that balances both aspects. As illustrated in Figure 2, Step 1 focuses on extracting key information from each sample, compressing it into a representation that captures its most informative components. This is achieved by maximizing the game-theoretic informativeness of each sample, which we measure using the Shapley Value (Shapley et al., 1953), a principled attribution method first introduced in game theory. In Step 2, we maximize the utility of each sample, which is critical for model training. This is done by measuring the gradient norm of each sample and selecting those with the highest values, ensuring that only the most valuable samples are retained. The main contributions of this work are summarized as follows:

1. We propose *Optimal Dataset Distillation* (Definition 4), which builds on the concepts of patch-wise Informativeness and sample-wise Utility for distilled datasets. This approach addresses the lack of interpretability in existing methods by providing a solid theoretical framework.

2. We introduce InfoUtil, a novel method balancing informativeness and utility in distilled dataset synthesis. It employs game-theoretic informativeness maximization via the Shapley Value and utility maximization to retain the most informative and valuable samples using the Gradient Norm.

3. InfoUtil demonstrates outstanding performance across various models and datasets. **For instance, our method yields a 16% improvement in performance over the previous state-of-the-art approach on the ImageNet-100, and a 6.1% improvement on ImageNet-1K.**

## 2 PRELIMINARIES

Given dataset $\mathcal{D} = \{(x_i, y_i)\}_{i=1}^n$, dataset distillation (DD) aims to synthesize a smaller dataset $\widetilde{\mathcal{D}} = \{(x_j, y_j)\}_{j=1}^m$ with $m \ll n$. The desired $\widetilde{\mathcal{D}}$ should enable a model to achieve comparable, even lossless, performance to one trained on $\mathcal{D}$, evaluated on a held-out test dataset $\mathcal{D}_{\text{test}}$. Specifically, for a model $f$ parameterized by $\theta$ trained with cross-entropy loss $\ell$, the condition is:

$$\min_{\widetilde{\mathcal{D}}} \sum_{(x,y) \in \mathcal{D}_{\text{test}}} |\ell(f_{\theta_{\mathcal{D}}}(x), y) - \ell(f_{\theta_{\widetilde{\mathcal{D}}}}(x), y)|, \tag{1}$$

where $\theta_{\mathcal{D}}$ denotes the fixed parameters trained on $\mathcal{D}$. Crucially, $\theta_{\widetilde{\mathcal{D}}}$ represents the parameters trained on the synthetic dataset $\widetilde{\mathcal{D}}$. Consequently, the term $\ell(f_{\theta_{\widetilde{\mathcal{D}}}}(x), y)$ depends on $\widetilde{\mathcal{D}}$ through the optimization trajectory of $\theta$.

This paper focuses on knowledge distillation-based DD methods, which recently showed superior performance (Yin et al., 2023; Sun et al., 2024; Shao et al., 2024). Here, $\mathcal{D}$'s information is first learned by a teacher model $f_{\theta_{\mathcal{D}}}$, which then synthesizes $\widetilde{\mathcal{D}}$. A notable work, RDED (Sun et al., 2024), uses random cropping to generate candidate patches, pruned via cross-entropy scoring. The final image contains multiple compressed images, each cropped and retained in prior steps. While RDED achieves high performance efficiently, it lacks principled guarantees. As Figure 1 shows, RDED's randomly selected patches often miss key ground truth category information.

## 3 METHOD

### 3.1 OPTIMAL DATASET DISTILLATION

To theoretically analyze the above problems, we first propose the following properties before formally defining the optimal dataset distillation mathematically.

**Definition 1 (Informativeness)** *Given an arbitrary sample $x \in \mathcal{D}$ and the compressed size $d' \ll d$, the informativeness of $x \in \mathbb{R}^d$ for the model $f_\theta$ is defined as:*

$$I(x; f_\theta) := -\big\| f_\theta(s \circ x) - f_\theta(x) \big\|, \tag{2}$$

*where $s \in \{0, 1\}^d$ and $|s| = d'$ is a $d$-dimensional binary mask to be optimized, $\circ$ is the Hadamard/element-wise product, and $s \circ x$ denotes the input $x$ with a mask $s$.*

The informativeness captures the key information for a given sample. Intuitively, maximizing the informativeness of a sample $x$ of a given compression size $d'$ can be regarded as learning the best informative mask vector $s$ that maximize the similarity of the performance between the original sample $x$ and the masked sample $s \circ x$.

Next, we introduce Gradient Flow, a key concept we use to define the Utility function.

**Definition 2 (Gradient Flow)** *Let $\ell_t$ be the cross-entropy loss for the model $\theta^{(t)}$ at iteration $t$. We define the gradient flow computed on a mini-batch $\mathcal{B}$ as:*

$$\dot{\ell}_t(f_{\theta^{(t)}}(x), y; \mathcal{B}) := \frac{\partial \ell_t(f_{\theta^{(t)}}(x), y)}{\partial t}. \tag{3}$$

The gradient flow $\dot{\ell}_t(f_{\theta^{(t)}}(x), y; \mathcal{B})$ represents the instantaneous rate of change of the loss for a specific example $(x, y)$ during training, providing a continuous-time approximation of training dynamics. Unlike discrete SGD updates, which introduce noise, gradient flow offers a smooth, analytical framework for quantifying data importance. By leveraging this, we assess the impact of removing a single data point $(x_i, y_i)$ and define a utility function below as a dataset pruning metric.

**Definition 3 (Utility)** *Let the gradient flow $\dot{\ell}_t$ be defined as in Definition 2. For a data point $(x_i, y_i)$ in dataset $\mathcal{D}$, let $\mathcal{B} \subseteq \mathcal{D}$ be the mini-batch at iteration $t$; define $\mathcal{B}_{\neg i} := \mathcal{B} \setminus \{(x_i, y_i)\}$. We measure the importance of $(x_i, y_i)$ by how much its removal changes the gradient flow over* all *relevant pairs:*

$$\mathcal{U}(x_i, y_i; f_{\theta^{(t)}}) := \max_{(x_j, y_j) \in \mathcal{D}} \left| \dot{\ell}_t(f_{\theta^{(t)}}(x_j), y_j; \mathcal{B}) - \dot{\ell}_t(f_{\theta^{(t)}}(x_j), y_j; \mathcal{B}_{\neg i}) \right|.$$

This utility definition captures the *worst-case* impact of removing a data point on gradient flow, ensuring it reflects data importance. By maximizing the change in $\dot{\ell}_t(f_{\theta^{(t)}}(x_j), y_j; \mathcal{B})$ over all $(x_j, y_j) \in \mathcal{D}$, it identifies points that most influence training dynamics. This aligns with dataset pruning by preserving critical samples while discarding those with minimal effect.

Based on Definition 1 and Definition 3, we propose the optimal dataset distillation in Definition 4:

**Definition 4 (Optimal Dataset Distillation)** *Let $f_\theta$ be the classifier model with parameter $\theta$ and $\mathcal{D}$ the original training dataset. Let $\mathcal{D}_{\text{test}}$ be the test dataset. Define $\mathcal{D}' \subseteq \mathcal{D}$ as a compressed subset, and $\widetilde{\mathcal{D}} \subseteq \mathcal{D}'$ as the final distilled dataset. Let $\mathcal{U}(x, y; f'_\theta)$ measure the utility of $f'_\theta$ on a test example $(x, y)$ defined in Definition 3. Let $I(x; f_\theta)$ measure the informativeness of original samples defined in Definition 1 and $s$ be the informative mask with compressed size $d'$. The goal is to find the optimal pruned dataset $\widetilde{\mathcal{D}}$ that maximizes both informativeness and utility on $\mathcal{D}_{\text{test}}$:*

$$\underset{\substack{\widetilde{\mathcal{D}} \subseteq \mathcal{D}' \\ |\widetilde{\mathcal{D}}| = m}}{\arg\max} \sum_{(x, y) \in \mathcal{D}_{\text{test}}} \mathcal{U}(x, y; f'_\theta), \quad \text{s.t.} \quad \mathcal{D}' = \left\{ x_i \circ s_i \; \middle| \; \underset{\substack{s_i \in \{0,1\}^d \\ |s_i| = d'}}{\arg\max} \; I(x_i; f_\theta) \right\}_{i=1}^n.$$

This formulation establishes the dataset distillation problem. The key challenge is then to define a rigorous utility function that effectively quantifies (i) the importance of each component within a sample for model prediction and also (ii) the importance of each sample for model training.

## 3.2 INFOUTIL

In this subsection, we introduce InfoUtil, built upon the *optimal dataset distillation* formulation in Definition 4. The pipeline has two main steps: (i) game-theoretic informativeness maximization and (ii) principled utility maximization. Detailed algorithm pseudocode is in Appendix A.

### 3.2.1 GAME-THEORETIC INFORMATIVENESS MAXIMIZATION

As in Definition 1, InfoUtil is to maximize the informativeness of each sample $x$ to obtain a compressed sample $s \circ x$, represented by a mask $s$. This task can be framed as a feature attribution problem (Zhou et al., 2016; Selvaraju et al., 2020; Binder et al., 2016; Shapley et al., 1953; Qin et al., 2023), where the model attributes decisions to input variables based on their importance.

Among attribution methods, the Shapley Value (Shapley et al., 1953) is regarded as a robust approach grounded in game theory. Specifically, given an input $x$ with $d$ input variables $x = [x^{(1)}, x^{(2)}, \dots, x^{(d)}]^\top$, we can view a deep neural network as a game with $d$ players $[d] := \{1, 2, \dots, d\}$. Each player $i$ corresponds to an input variable $x^{(i)}$. Thus, the task of fairly assigning the reward in the game translates to fairly estimating attributions of input variables in the deep neural network $f$. Formally, the Shapley value $\phi$ can be defined as:

$$\phi_f(x^{(i)}) = \frac{1}{d} \sum_{s: s_i = 0} \binom{d-1}{\mathbf{1}^\top s} \left( f(x \circ (s + e_i)) - f(x \circ s) \right), \tag{4}$$

where $e_i \in \mathbb{R}^d$ denotes the vector with a one in the $i$-th position but zeros in the rest positions. Notably, the Shapley Value is renowned for satisfying four key axioms (Young, 1985):

For detailed technical derivations, including the complete proof, please refer to Appendix D.

**Axiom 1 (Linearity. Proof in Appendix D.1)** *If two games can be merged into a new game, then the Shapley Values in the two original games can also be merged. Formally, if $f_{\mathrm{merged}} = f_1 + f_2$, then $\phi_{f_{\mathrm{merged}}}(x^{(i)}) = \phi_{f_1}(x^{(i)}) + \phi_{f_2}(x^{(i)}), \forall i \in [d]$.*

**Axiom 2 (Dummy. Proof in Appendix D.2)** *A dummy player $i$ is a player that has no interactions with other players in the game $f$. Formally, if $\forall s : s_i = 0$, $f(x \circ (s + e_i)) = f(x \circ s) + f(x \circ e_i)$. Then, the dummy player's Shapley Value is computed as $f(x \circ e_i)$.*

**Axiom 3 (Symmetry. Proof in Appendix D.3)** *If two players contribute equally in every case, then their Shapley values in the game $f$ will be equal. Formally, if $\forall s : s_i = s_j = 0$, $f(x \circ (s + e_i)) = f(x \circ (s + e_j))$, then $\phi_f(x^{(i)}) = \phi_f(x^{(j)})$.*

**Axiom 4 (Efficiency. Proof in Appendix D.4)** *The total reward of the game $f$ is equal to the sum of the Shapley values of all players. Formally, $f(x) - f(\mathbf{0}) = \sum_{i \in [d]} \phi_f(x^{(i)})$.*

The Shapley value is the unique attribution method that satisfies the four key axioms (Young, 1985). However, directly computing the Shapley value is computationally expensive in practice. For instance, calculating the Shapley value for an image with $4 \times 4$ patches requires $2^{16}$ inferences, assuming each patch is a player. To address this issue, prior works (Charnes et al., 1988; Lundberg & Lee, 2017) have proposed using kernel-based estimation of the Shapley value, as follows:

$$\phi = \arg\min_{\phi} \quad \mathbb{E}_{s \sim q(s)}\left[\left(f(x \circ s) - f(\mathbf{0}) - s^\top \phi\right)^2\right], \quad \text{s.t.} \quad \mathbf{1}^\top \phi = f(x) - f(\mathbf{0}), \quad (5)$$

where $q(s) = (d-1)/\left(\binom{d}{\mathbf{1}^\top s}(\mathbf{1}^\top s)(d - \mathbf{1}^\top s)\right), \forall 1 < \mathbf{1}^\top s < d$ denotes the Shapley Kernel. We follow KernelShap (Lundberg & Lee, 2017) to achieve fast estimation of the Shapley value based on Eq. (5), making it possible to be adept in practice.

After obtaining the Shapley value $\phi_f(x^{(i)})$ of each sample $x^{(i)}$, we apply average pooling of the Shapley value map $\phi_f(x) = [\phi_f(x^{(1)}), \phi_f(x^{(2)}), \ldots, \phi_f(x^{(d)})]$ to obtain the most informative region inside a image. This step would generate a $d' < d$ size compressed image (*e.g.*, $d' = d/4$) with the maximized informativeness, resulting a compressed dataset with $n$ compressed samples $\mathcal{D}'$.

**Diversity control**. The Shapley value attribution typically identifies only the most informative patch. To introduce diversity in the patch selection process, we incorporate random noise $\varepsilon \sim (0, \sigma^2)$, where $\sigma$ is the standard deviation fixed. Specifically, the random noise is employed on the average pooled attribution heatmap, resulting in diverse informative patches considered in the next phase.

### 3.2.2 PRINCIPLED UTILITY MAXIMIZATION

After obtaining the compressed dataset, the next step is selecting samples to maximize dataset utility. Computing utility (Definition 3) is challenging, as it requires training models with and without each sample $x$ to assess its utility. We show the utility function can be upper-bounded by the gradient norm (Theorem 1), simplifying computation. We now define the gradient norm.

**Definition 5 (Gradient Norm)** *The gradient norm of a training example $(x, y)$ for model $f$ parameterized by $\theta^{(t)}$ at time $t$ is denoted as*

$$\|\nabla_{\theta^{(t)}} \ell_t(f_{\theta^{(t)}}(x), y)\|.$$

Given the definition of Gradient Norm, we then show that Utility can be upper bounded by the gradient norm through detailed analysis here.

**Theorem 1 (Utility is bounded by Gradient Norm. Proof in Appendix E)** *Let the utility function $\mathcal{U}$ be defined as in Definition 3. Then there exists a constant $c > 0$ such that*

$$\mathcal{U}(x_i, y_i; f_{\theta^{(t)}}) \leq c\|\nabla_{\theta^{(t)}} \ell_t(f_{\theta^{(t)}}(x_i), y_i)\|.$$

*Proof of Theorem 1.* For detailed technical derivations, including the complete proof of Theorem 1 and auxiliary lemmas, please refer to the supplementary materials. The full proof includes step-by-step expansions of gradient flow decompositions, rigorous bounds under SGD updates, and verification of assumptions underlying the utility-gradient norm relationship.

Table 1: Performance comparison between InfoUtil and SOTA methods on seven datasets. We evaluate dataset distillation using ResNet-18, ResNet-101, and ConvNet, reporting top-1 accuracy (%).Datasets were distilled with ResNet-18 and ConvNet, then evaluated on matching architectures. Additionally, datasets distilled by ResNet-18 were also evaluated with ResNet-101.

| Dataset | IPC | ResNet-18 | | | ResNet-101 | | | ConvNet | | | | | |
|---|---|---|---|---|---|---|---|---|---|---|---|---|---|
| | | SRe2L | RDED | InfoUtil | SRe2L | RDED | InfoUtil | MTT | IDM | TESLA | DATM | RDED | InfoUtil |
| CIFAR-10 | 1 | 16.6±0.9 | 22.9±0.4 | **25.3**±0.4 | 13.7±0.2 | 18.7±0.1 | **19.6**±0.6 | 46.3±0.8 | 45.6±0.7 | **48.5**±0.8 | 46.9±0.5 | 23.5±0.3 | 28.5±1.4 |
| | 10 | 29.3±0.5 | 37.1±0.3 | **53.8**±0.1 | 24.3±0.6 | 33.7±0.3 | **38.4**±1.0 | 65.3±0.7 | 58.6±0.1 | 66.4±0.8 | **66.8**±0.2 | 50.2±0.3 | 54.1±0.5 |
| | 50 | 45.0±0.7 | 62.1±0.1 | **71.0**±0.8 | 34.9±0.1 | 51.6±0.4 | **67.1**±0.5 | 71.6±0.2 | 67.5±0.1 | 72.6±0.7 | **76.1**±0.3 | 68.4±0.1 | 69.8±0.1 |
| CIFAR-100 | 1 | 6.6±0.2 | 11.0±0.3 | **22.9**±0.4 | 6.2±0.0 | 10.8±0.1 | **16.5**±0.5 | 24.3±0.3 | 20.1±0.3 | 24.8±0.5 | 27.9±0.2 | 19.6±0.3 | **33.1**±0.3 |
| | 10 | 27.0±0.4 | 42.6±0.2 | **47.5**±0.7 | 30.7±0.3 | 41.1±0.2 | **41.9**±0.6 | 40.1±0.4 | 45.1±0.1 | 41.7±0.3 | 47.2±0.4 | 48.1±0.3 | **50.5**±0.2 |
| | 50 | 50.2±0.4 | 62.6±0.1 | **64.7**±0.2 | 56.9±0.1 | 63.4±0.3 | **66.0**±0.2 | 47.7±0.2 | 50.0±0.2 | 47.9±0.3 | 55.0±0.2 | 57.0±0.1 | **57.8**±0.2 |
| ImageNette | 1 | 19.1±1.1 | 35.8±1.0 | **43.8**±0.7 | 15.8±0.6 | 25.1±2.7 | **28.2**±0.5 | 47.7±0.9 | - | - | - | 33.8±0.8 | 42.3±0.7 |
| | 10 | 29.4±3.0 | 61.4±0.4 | **68.6**±0.6 | 23.4±0.8 | 54.0±0.4 | **59.8**±1.1 | 63.0±1.3 | - | - | - | 63.2±0.7 | **66.6**±0.4 |
| | 50 | 40.9±0.3 | 80.4±0.4 | **86.2**±0.6 | 36.5±0.7 | 75.0±1.2 | **82.4**±0.3 | - | - | - | - | 83.8±0.2 | **84.4**±0.6 |
| ImageWoof | 1 | 13.3±0.5 | 20.8±1.2 | **25.0**±0.8 | 13.4±0.1 | 19.6±1.8 | **20.2**±0.4 | 28.6±0.8 | - | - | - | 18.5±0.9 | 22.8±0.4 |
| | 10 | 20.2±0.2 | 38.5±2.1 | **51.4**±2.5 | 17.7±0.9 | 31.3±1.3 | **42.6**±1.2 | 35.8±1.8 | - | - | - | 40.6±2.0 | **43.8**±1.3 |
| | 50 | 23.3±0.3 | 68.5±0.7 | **69.6**±0.8 | 21.2±0.2 | 59.1±0.7 | **67.2**±0.8 | - | - | - | - | 61.5±0.3 | **62.6**±0.4 |
| Tiny-ImageNet | 1 | 2.6±0.1 | 9.7±0.4 | **17.0**±1.3 | 1.9±0.1 | 3.8±0.1 | **11.9**±0.6 | 8.8±0.3 | 10.1±0.2 | - | 17.1±0.3 | 12.0±0.1 | **19.6**±0.5 |
| | 10 | 16.1±0.2 | 41.9±0.2 | **45.6**±0.3 | 14.6±1.1 | 22.9±3.3 | **34.4**±0.2 | 23.2±0.2 | 21.9±0.3 | - | 31.1±0.3 | 39.6±0.1 | **40.2**±0.3 |
| | 50 | 41.1±0.4 | 58.2±0.1 | **58.5**±0.3 | 42.5±0.2 | 41.2±0.4 | **54.7**±0.3 | 28.0±0.3 | 27.7±0.3 | - | 39.7±0.3 | 47.6±0.2 | **48.0**±0.5 |
| ImageNet-100 | 1 | 3.0±0.3 | 8.1±0.3 | **15.7**±0.2 | 2.1±0.1 | 6.1±0.8 | **11.4**±0.2 | - | 11.2±0.5 | - | - | 7.1±0.2 | 15.0±0.8 |
| | 10 | 9.5±0.4 | 36.0±0.3 | **50.5**±0.4 | 6.4±0.1 | 33.9±0.1 | **49.9**±0.4 | - | 17.1±0.6 | - | - | 29.6±0.1 | **42.2**±0.7 |
| | 50 | 27.0±0.4 | 61.6±0.1 | **68.3**±0.4 | 25.7±0.3 | 66.0±0.6 | **69.7**±0.4 | - | 26.3±0.4 | - | - | 50.2±0.2 | **60.8**±0.9 |
| ImageNet-1K | 1 | 0.1±0.1 | 6.6±0.2 | **12.8**±0.7 | 0.6±0.1 | 5.9±0.4 | **6.8**±0.7 | - | - | 7.7±0.2 | - | 6.4±0.1 | 6.6±0.3 |
| | 10 | 21.3±0.6 | 42.0±0.1 | **44.2**±0.4 | 30.9±0.1 | 48.3±1.0 | **51.4**±0.3 | - | - | 17.8±1.3 | - | 20.4±0.1 | **21.5**±0.3 |
| | 50 | 46.8±0.2 | 56.5±0.1 | **58.0**±0.3 | 60.8±0.5 | 61.2±0.4 | **63.8**±0.6 | - | - | 27.9±1.2 | - | 38.4±0.2 | **40.2**±0.4 |

Given Theorem 1, we can efficiently calculate the utility of each sample using the upper bound of the gradient norm. Then, we can directly select the most influential samples with the highest gradient norms to maximize utility. Specifically, we employ gradient norm scoring for all compressed samples in $\mathcal{D}'$ with size $n$, and selected samples with top norm scores, resulting $\widetilde{\mathcal{D}}$ with size $m \ll n$.

**Image Reconstruction.** Following prior works (Yin et al., 2023; Sun et al., 2024; Shao et al., 2024), we reconstruct normal-sized images by combining compressed samples. Low-resolution datasets use a single image per category, while high-resolution datasets merge four $1/4$-resolution images from the same category into one full-size image. For soft label generation, patch-specific logits are assigned by resizing the compressed samples. Inspired by (Qin et al., 2024; Wang et al., 2024b), intermediate checkpoints of a pretrained model are used to balance discriminativity and diversity, improving performance. Further details are in Section 6.

## 4 EXPERIMENTS

### 4.1 EXPERIMENTAL SETTINGS

**Datasets and network architectures.** We evaluated our approach using widely recognized datasets. For lower-resolution datasets, we employed CIFAR-10 and CIFAR-100 (Krizhevsky et al., 2009) ($32 \times 32$) and Tiny-ImageNet (Deng et al., 2009) ($64 \times 64$). For higher-resolution experiments, we used ImageNet-1K (Deng et al., 2009) ($224 \times 224$) along with three commonly used ImageNet subsets: ImageNette, ImageWoof, and ImageNet-100 (all at $224 \times 224$). In line with previous works on dataset distillation, we adept the following backbone architectures: ConvNet (Liu et al., 2022), ResNet-18, 50, 101 (He et al., 2016), MobileNet-V2 (Howard et al., 2019), VGG-11 (Simonyan & Zisserman, 2014), and Swin-V2-Tiny (Liu et al., 2021). Specifically, dataset distillation is performed using a 3-layer ConvNet for CIFAR-10/100, a 4-layer ConvNet for Tiny-ImageNet and ImageNet-1K, a 5-layer ConvNet for ImageWoof and ImageNette, and a 6-layer ConvNet for ImageNet-100.

**Baseline methods.** Following previous studies, we assessed the quality of the condensed datasets by training neural networks from scratch using them. We reported the resulting test accuracies on the actual validation sets. Baseline include trajectory-matching approaches such as MTT (Cazenavette

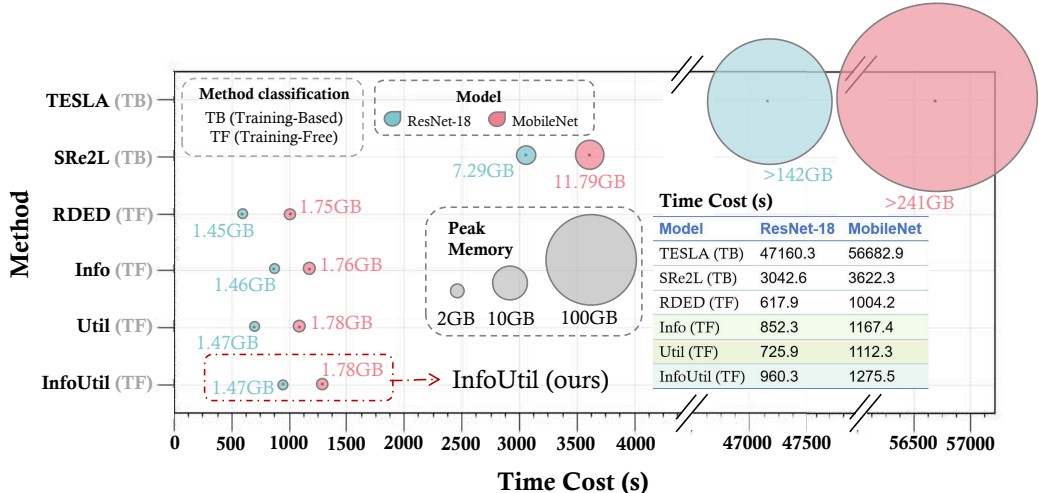

Figure 3: Performance comparison on ResNet-18 and MobileNet. (a) Time cost in seconds (lower is better): "TB" denotes training-based methods (TESLA and SRe2L fall into this category); "TF" denotes training-free methods (others belong to this type). (b) Peak memory in GB (lower is better): InfoUtil performs competitively with far lower costs than training-based methods. "Info" denotes Informativeness only, while "Util" denotes Utility only.

et al., 2022), TESLA (Cui et al., 2023), and DATM (Guo et al., 2023), and distribution-matching methods like IDM (Zhao et al., 2023). For our primary comparison, we also include SOTA knowledge distillation-based methods, SRe2L (Yin et al., 2023) and RDED (Sun et al., 2024).

**Implementation details of InfoUtil.** Our setup follows RDED, using pretrained networks for dataset synthesis. For small IPC, we adopt the approach in (Qin et al., 2024), extracting training-stage soft labels to capture rich semantics. For larger IPC, fully converged networks from RDED are used. Details are in Appendix A. For low-resolution datasets, one synthetic image per class is used, while high-resolution datasets use four per class. The 300-image subset matches RDED's configuration. As in Table 1, AutoAug (Cubuk et al., 2018) is applied to enhance synthetic dataset performance. All experiments ran on a single NVIDIA A100 GPU.

## 4.2 MAIN RESULTS

We verified InfoUtil's effectiveness on benchmark datasets across image-per-class (IPC) settings.

**Higher-resolution datasets.** We benchmarked InfoUtil against state-of-the-art methods on higher-resolution datasets like ImageNet-1K and its subsets. As Table 1 shows, InfoUtil achieves superior or comparable performance across IPC settings. Notably, on ImageNet-100 (ResNet-101, IPC=10), it outperforms RDED (Sun et al., 2024) by 16% in accuracy; on ImageWoof (ResNet-18, IPC=10), it gains 12.9% over RDED. Moreover, on ImageNet-1K (ResNet-18, IPC=1), InfoUtil surpasses RDED by 6.1%, highlighting its effectiveness in small IPC scenarios.

**CIFAR-10/100 and Tiny-ImageNet.** We evaluated InfoUtil on lower-resolution datasets with additional experiments on CIFAR-10/100 and Tiny-ImageNet. Our method continues to show superior performance across most scenarios, highlighting robustness and generalizability of InfoUtil. Specifically, as in Table 1, on Tiny-ImageNet, using ResNet-101 at IPC = 50 yields a 13.5% improvement; on CIFAR-10, ResNet-18 at IPC = 10 obtains a 16.7% improvement.

**Cross-architecture generalization.** We evaluated InfoUtil's cross-architecture generalization across ResNet-18/50 (He et al., 2016), VGG-11 (Simonyan & Zisserman, 2014), MobileNet-V2 (Howard et al., 2019), and Swin-V2-Tiny (Liu et al., 2021). Table 2 shows InfoUtil outperforms SOTA (SRe2L, RDED) by 10% in the VGG-11 (teacher) vs. Swin-V2-Tiny (student) setting, confirming versatility. Further validation with baselines SCDD, G-VBSM (structural regularization) and D3S (data efficiency) on ImageNet-1K across ResNet-18/101 (Table 3) shows InfoUtil consistently outperforms SRe2L/RDED and these baselines across all IPC settings.

**Efficiency Analysis.** We carefully measured InfoUtil's runtime and GPU usage on a single NVIDIA A100. (i) It is highly efficient: time is **50× lower** and memory **100× smaller** than TESLA across all distillation stages (Figure 3). (ii) For large-scale datasets like ImageNet-21K, distillation com-

Table 2: Cross-architecture performance (%) on ImageNet-1K (IPC=10). Using ResNet-18/50, VGG-11, MobileNet-V2, and Swin-V2-Tiny as teachers; ResNet-18, MobileNet-V2, and Swin-V2-Tiny as students.

| Squeezed\Evaluation | | ResNet-18 | MobileNet-V2 | Swin-V2-Tiny |
|---|---|---|---|---|
| ResNet-18 | SRe2L | 21.7±0.6 | 15.4±0.2 | - |
| | RDED | 42.3±0.6 | **40.4**±0.1 | 17.2±0.2 |
| | **InfoUtil** | **44.8**±0.4 | 37.1±0.5 | **19.8**±0.4 |
| ResNet-50 | SRe2L | - | - | - |
| | RDED | 33.9±0.5 | 26.0±0.3 | **17.3**±0.2 |
| | **InfoUtil** | **34.7**±1.4 | **28.1**±0.6 | 15.6±0.4 |
| MobileNet-V2 | SRe2L | 19.7±0.1 | 10.2±2.6 | - |
| | RDED | 34.4±0.2 | 33.8±0.8 | 11.8±0.3 |
| | **InfoUtil** | **39.2**±0.3 | **35.5**±0.5 | **20.6**±0.2 |
| VGG-11 | SRe2L | 16.5±0.1 | 10.6±0.1 | - |
| | RDED | 22.7±0.1 | 21.6±0.2 | 7.8±0.1 |
| | **InfoUtil** | **35.1**±0.3 | **31.6**±0.1 | **17.8**±0.4 |
| Swin-V2-Tiny | SRe2L | 9.6±0.3 | 7.4±0.1 | - |
| | RDED | 17.8±0.1 | 18.1±0.2 | 12.1±0.2 |
| | **InfoUtil** | **18.4**±0.4 | **19.7**±0.4 | **16.4**±0.3 |

Table 3: Cross-architecture comparison of InfoUtil with additional baselines on ImageNet-1K. Results are shown across ResNet-18 and ResNet-101 architectures under varied IPC settings.

| Model | ResNet-18 | | | ResNet-101 | | |
|---|---|---|---|---|---|---|
| IPC | 1 | 10 | 50 | 1 | 10 | 50 |
| SRe2L | 0.1±0.1 | 21.3±0.6 | 46.8±0.2 | 0.6±0.1 | 30.9±0.1 | 60.8±0.5 |
| SCDD | - | 32.1±0.2 | 53.1±0.1 | - | 39.6±0.4 | 61.0±0.3 |
| G-VBSM | - | 31.4±0.5 | 51.8±0.4 | - | 38.2±0.4 | 61.0±0.4 |
| D3S | 5.3±0.1 | 37.2±0.3 | 50.3±0.3 | 3.2±0.8 | 42.3±1.7 | 60.6±0.2 |
| RDED | 6.6±0.2 | 42.0±0.1 | 56.5±0.1 | 5.9±0.4 | 48.3±1.0 | 61.2±0.4 |
| **InfoUtil** | **12.7**±0.7 | **44.2**±0.4 | **58.0**±0.3 | **6.8**±0.7 | **51.4**±0.3 | **63.7**±0.6 |

Table 4: Comparison of downstream tasks for distilled samples in 5-step continual learning. Higher values indicate better performance.

| Method | Stage 1 | Stage 2 | Stage 3 | Stage 4 | Stage 5 |
|---|---|---|---|---|---|
| RDED | 0.5153 | 0.2918 | 0.201 | 0.1967 | 0.2191 |
| **InfoUtil** | **0.6560** | **0.5659** | **0.4927** | **0.4617** | **0.4739** |

Table 5: Comparison with baseline methods under large IPC settings. We used ResNet-18 for dataset synthesis on Tiny-ImageNet and ImageNet-1K, and evaluated on ResNet-18 and ResNet-50 models. Note that TESLA (Cui et al., 2023) used the downsampled ImageNet-1K dataset.

| Dataset | IPC | TESLA (R18) | SRe2L (R18) | RDED (R18) | InfoUtil (R18) | SRe2L (R50) | InfoUtil (R50) |
|---|---|---|---|---|---|---|---|
| Tiny-ImageNet | 50 | - | 41.1±0.4 | 58.2±0.1 | **58.5**±0.3 | 42.2±0.5 | **48.3**±0.4 |
| | 100 | - | 49.7±0.3 | 59.9±0.4 | **60.6**±0.5 | 51.2±0.4 | **53.7**±0.4 |
| | 200 | - | 51.2±0.6 | 61.5±0.3 | **62.0**±0.3 | - | **58.0**±0.3 |
| ImageNet-1K | 10 | 17.8±1.3 | 21.3±0.6 | 42.0±0.1 | **43.5**±0.4 | 28.4±0.1 | **48.0**±0.5 |
| | 50 | 27.9±1.2 | 46.8±0.2 | 56.5±0.1 | **57.6**±0.3 | 55.6±0.3 | **63.1**±0.4 |
| | 100 | - | 52.8±0.3 | 58.2±0.6 | **58.8**±0.4 | 61.0±0.4 | **65.5**±0.5 |
| | 200 | - | 57.0±0.4 | 62.5±0.8 | **63.4**±0.3 | 64.6±0.3 | **68.0**±0.4 |

pletes in just 5.83 hours. This combination of remarkable efficiency and strong performance makes InfoUtil a practical, scalable solution for modern dataset distillation.

**Performance on large IPC settings.** We tested Tiny-ImageNet and ImageNet-1K under large IPC scenarios, comparing with bi-level Tesla (Cui et al., 2023) and uni-level SRe2L (Yin et al., 2023), RDED (Sun et al., 2024). Table 5 shows our method significantly outperforms existing SOTA in large IPC cases, demonstrating strong scalability and superior performance. For IPC=200 on ImageNet-1K, we used full images (not 2×2 cropped patches as prior work) to mitigate imbalance (following (Sun et al., 2024)); image count before scoring was 600 instead of 300.

**Downstream tasks of distilled samples.** We explored the effectiveness of distilled samples in downstream tasks via experiments on ImageNette (50 IPC) with 5-step continual learning, where new classes are incrementally introduced at each stage. To ensure the robustness of results, experiments were repeated 5 times with varied class orders. As shown in Table 4, our method (InfoUtil) consistently surpasses the SOTA method RDED across all stages.

**Visualization.** InfoUtil shows significant improvements in visual quality over existing methods. First, vs. optimization-based methods like SRe2L (Yin et al., 2023), it produces more realistic representations by preserving intricate details and maintaining natural color fidelity. Second, vs. optimization-free methods like RDED (Sun et al., 2024), InfoUtil is more interpretable and principled, effectively capturing key informative semantic content while minimizing focus on irrelevant regions. Due to space constraints, visualization images are provided in Appendix F.

## 4.3 ABLATION STUDIES

To analyze the individual contributions of InfoUtil's components, we conducted comprehensive ablation studies comparing three configurations: (1) the baseline RDED method (Rand. Crop + Loss Scoring), (2) Utility Maximization alone (GradN Scoring), and (3) the complete InfoUtil (GradN Scoring + Attri. Cropping). Results across multiple datasets (ImageWoof, ImageNette, ImageNet-1K) with varying IPC values are presented in Table 6.

| GradNorm Scoring | Attribution Cropping | ImageWoof IPC=1 | ImageWoof IPC=50 | ImageNette IPC=50 | ImageNet-1K IPC=10 |
|---|---|---|---|---|---|
| ✗ | ✓ | 38.5 | 68.5 | 80.4 | 42.0 |
| ✓ | ✗ | 43.6 | 68.8 | 85.0 | 43.5 |
| ✓ | ✓ | 45.2 | 69.6 | 86.2 | 44.2 |

Table 6: Ablation study of InfoUtil components' impact on image classification. Top-1 accuracy (%) on ResNet-18 across datasets are reported.

• **Effect of Utility Maximization**. Replacing Loss Scoring with GradN Scoring while maintaining random cropping brings significant performance improvements. As shown in Table 6, Utility Maximization alone achieves a 4.6% performance boost on ImageNette (IPC=50, from 80.4% to 85.0%) and a 1.5% improvement on ImageNet-1K (IPC=10, from 42.0% to 43.5%). These results demonstrate that gradient norm-based scoring plays a crucial role in selecting more informative samples.

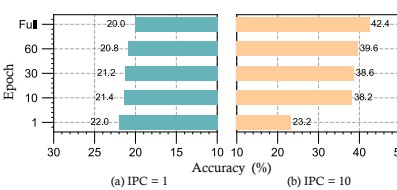

Figure 4: Analysis of teacher networks for soft label generation. ConvNet performance on ImageWoof using labels from five training stages (IPC=1/10). "Full" denotes pre-trained teacher. (a) IPC=1: Early high-entropy labels beat full model, aiding low-data scenarios. (b) IPC=10: Full model's low-entropy labels excel in data-rich conditions.

• **Effect of Combined Components**. The integration of both Utility Maximization and Informativeness Maximization through Attri. Cropping yields the best performance. InfoUtil achieves additional gains of 1.2% on ImageNette (reaching 86.2%) and 0.7% on ImageNet-1K (reaching 44.2%) compared to using Utility Maximization alone. This synergistic combination demonstrates that attribute-guided cropping effectively captures the most discriminative regions while gradient-based scoring ensures the selection of pedagogically valuable samples, together producing high-quality synthetic data that consistently outperforms the baseline across all experimental settings.

• **Effect of Noise Injection**. We investigate the role of noise injection in attribution-guided patch selection, which is essential for maintaining synthetic data diversity. Without noise, the selection process becomes deterministic and greedy, leading to redundant synthesis that hinders model generalization. As shown in Table 7, removing noise ("w/o Noise") results in significant performance degradation across all settings. Notably, the performance gap exceeds 15% at higher IPC (e.g., 86.2% vs. 70.6% on ImageNette at IPC=50), confirming that noise-induced diversity is critical for effective feature space coverage. Even at IPC=1, the consistent gains suggest that noise helps identify robust central features rather than brittle local maxima in the attribution map.

Table 7: Ablation Study on Noise Injection: Top-1 Accuracy across various settings.

| Dataset IPC | ImageNette Standard | ImageNette w/o Noise | ImageWoof Standard | ImageWoof w/o Noise | ImageNet-100 Standard | ImageNet-100 w/o Noise | ImageNet-1K Standard | ImageNet-1K w/o Noise |
|---|---|---|---|---|---|---|---|---|
| 1 | 43.8 | 35.4 | **25.0** | 23.2 | **15.7** | 12.6 | **12.8** | 9.6 |
| 10 | 68.6 | 59.8 | **51.4** | 40.0 | **50.5** | 43.8 | **44.2** | 38.5 |
| 50 | **86.2** | 70.6 | **69.6** | 59.4 | **68.3** | 56.3 | **58.0** | 48.3 |

• **Attribution Method: Shapley vs. Grad-CAM**. We justify the selection of Shapley Value over Grad-CAM by comparing their theoretical rigor and empirical performance. Unlike heuristic gradient-based methods like Grad-CAM, which often suffer from gradient saturation, the Shapley Value is the unique attribution method satisfying fundamental axioms such as Efficiency and Symmetry. This theoretical robustness ensures a fair distribution of "Informativeness," accurately capturing the marginal contribution of each patch. Empirical results in Table 8 confirm this advantage, with Shapley-based selection consistently outperforming Grad-CAM on ImageNet-1K. Notably, at IPC=10, Shapley achieves 43.88%, surpassing Grad-CAM by 13.49%, demonstrating its superior ability to identify semantically robust patches for effective distillation.

## 5 RELATED WORK

**Dataset Distillation.** Dataset Distillation aims to reduce a large dataset into a smaller one. Current methods can be categorized into two main approaches: *i.e.*, matching-based methods (Zhao et al., 2021; Lee et al., 2022; Zhao & Bilen, 2021; Wang et al., 2024a; Cazenavette et al., 2022; Cui et al.,

Table 8: Empirical Comparison of Attribution Methods: Shapley Value vs. Grad-CAM on ImageNet-1K (ResNet-18 Student Model).

| Model | Dataset | IPC | Grad-CAM | Shapley (Ours) |
|-------|---------|-----|----------|----------------|
| ResNet-18 | ImageNet-1K | 1 | 4.418 | **7.154** |
|  |  | 10 | 30.394 | **43.880** |
|  |  | 50 | 52.610 | **56.920** |

2023; Guo et al., 2023; Zhao & Bilen, 2022; Kim et al., 2022; Du et al., 2023; Zhou et al., 2022; Wang et al., 2025c; Min et al., 2026), and knowledge-distillation-based methods (Yin et al., 2023; Shao et al., 2024; Sun et al., 2024; Wang et al., 2025a). Matching-based methods are typically formulated as bi-level optimization problems but struggle with the trade-off between efficiency and the quality of the distilled dataset. In contrast, knowledge-distillation-based methods decouple the problem into a two-step process but often lack theoretical interpretability. Therefore, a deeper investigation is needed to formalize knowledge-distillation-based methods in a principled manner to ensure their reliability in practical scenarios with theoretical support, which we address in this paper.

**Attribution Methods in Explainable AI.** Attribution methods are essential for post-hoc explanations of black-box models, revealing each input variable's contribution to the final prediction. Among them, the Shapley Value is considered a principled tool due to its key axioms: *i.e.*, *linearity*, *dummy*, *symmetry*, and *efficiency* (Shapley et al., 1953; Young, 1985). To reduce the computational burden, KernelShap (Lundberg & Lee, 2017) was introduced to efficiently approximate the Shapley Value using Linear LIME (Ribeiro et al., 2016) or neural network (Wang et al., 2025b). However, since none of the previous works have explored the application of attribution methods in dataset distillation, there is an opportunity to develop attribution-based approaches for extracting key information for dataset distillation.

## 6 DISCUSSION

Soft labels encode richer probabilistic supervision in dataset distillation. Prior works (Guo et al., 2023; Yin et al., 2023; Wang et al., 2024b; Qin et al., 2024; Sun et al., 2024) show they capture inter-class relationships. (Qin et al., 2024) finds early high-entropy labels help low-data regimes, while late low-entropy labels suit data-rich settings. (Wang et al., 2024b) notes effective labels balance diversity and discriminability. However, these focus on matching-based distillation, leaving knowledge-distillation-based DD with soft labels unexplored.

To investigate this further, we explored the effectiveness of teacher model for soft label generation using ConvNet on ImageWoof. For small IPC settings, we extracted soft labels from models at an intermediate training stage (10-th epoch), leveraging the high-entropy, diverse information characteristic of early epochs. In contrast, for large IPC settings, we used fully pretrained networks from RDED, leveraging the low-entropy, precise labels typical of later training phases.

Our findings, as it shown in Figure 4, clearly highlight the effectiveness of this strategy. In small IPC scenarios (*e.g.*, IPC = 1), synthetic images with soft labels generated with models at 10-th epoch outperformed those from pretrained networks, emphasizing the importance of rich label information when limited data are provided. Conversely, in larger IPC scenarios (*e.g.*, IPC = 10 or IPC = 50), labels from fully pretrained networks yielded superior results.

## 7 CONCLUSION

In this paper, we present a principled approach to dataset distillation, grounded in a rigorous theoretical framework for modeling optimal distillation. We introduce *Informativeness* and *Utility*, capturing, the critical information within a sample and essential samples for effective training. Building on these, we propose InfoUtil, a framework that synergistically combines game-theoretic informativeness maximization with principled utility maximization. Specifically, InfoUtil leverages Shapley value attribution to extract informative features and employs gradient norm-based optimization to select samples optimized for utility. InfoUtil demonstrates superior performance in dataset distillation and cross-architecture generalization. Future work includes extending InfoUtil to more complex and diverse datasets, focusing on scalability and robustness in real-world applications.

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

---

**Algorithm 1** InfoUtil Pipeline

---

**Input:** original dataset $\mathcal{D}$, pre-trained teacher model $f_{\theta_{\mathcal{D}}}$, teacher model at early $t$-th epoch $f_{\theta_t}$, compressed size $d'$, noise variance $\sigma$, distilled dataset size $m$, number of patches $k$.

**for** each class $c$ in $\mathcal{D}$ **do**

    $\mathcal{D}_c = \{(x_i, y_i) \in \mathcal{D} \mid y_i = c\}$

    // **Stage 1: Informativeness Maximization**

    **for** $(x_i, y_i) \in \mathcal{D}_c$ **do**

        Compute $\phi_f(x_i)$ using $f_{\theta_{\mathcal{D}}}$

        Apply average pooling to $\phi_f(x_i)$ to obtain a pooled heatmap

        Add noise $\varepsilon \sim (0, \sigma^2)$ to the pooled heatmap

        Extract $\xi_i$ of size $d'$ from $x_i$ based on the highest heatmap value

    **end for**

    $\mathcal{D}'_c = \{(\xi_i, y_i)\}$

    // **Stage 2: Utility Maximization**

    **for** $(\xi_i, y_i) \in \mathcal{D}'_c$ **do**

        Compute $g_i = \|\nabla_\theta \ell(f_{\theta_{\mathcal{D}}}(\xi_i), y_i)\|$

    **end for**

    Select top-$k \times$ IPC samples $\{\xi_{i1}, \ldots, \xi_{i, \text{IPC} \times k}\}$ by $g_i$

    **for** $j = 1$ to IPC **do**

        Combine $\xi_{i,(j-1)\times k+1}$ to $\xi_{i,j\times k}$ into $x_j$

        For each $\xi_{ik}$ in $x_j$, set $\widetilde{y_{jk}} = f_{\theta_t}(\xi_{ik})$

        $\widetilde{y_j} = [\widetilde{y_{j1}}, \ldots, \widetilde{y_{jk}}]$

        $\widetilde{\mathcal{D}} = \widetilde{\mathcal{D}} \cup \{(x_j, y_j)\}$

    **end for**

**end for**

**Output:** Distilled dataset $\widetilde{\mathcal{D}}$

---

## A DETAILED IMPLEMENTATION

In this section, we detail the implementation specifics of InfoUtil, including the computation of informativeness, tuning settings of teacher models, and provide the corresponding pseudocode in the Algorithm 1.

### A.1 COMPUTATION OF INFORMATIVENESS

In our implementation of InfoUtil, we leveraged the PyTorch framework together with the Captum package to compute Shapley values. Captum provides a robust and flexible interface for model interpretability, allowing us to quantitatively assess the contributions of individual features to the model's predictions. By utilizing Captum's KernalShap[1] method, we could accurately determine the importance of each feature within a sample, which in turn guides the data refinement process during dataset distillation. Moreover, in the first four cropping, we injected Gaussian noise drawn from the normal distribution $\mathcal{N}(0, \sigma^2)$, where $\sigma$ is defined as the product of the overall standard deviation of the Shapley values after average pooling and a hyperparameter $\alpha$ that controls the noise intensity. In our experiments, we set the kernal size to $2 \times 2$ with stride = 1, and the hyperparameter $\alpha = 2$. The final (5th) cropping maintained the original Shapley values. This approach effectively reduced the probability of repeatedly cropping the same location.

Besides, in most scenarios, we divided each original image into a $4 \times 4$ grid of patches, computed the Shapley value for each individual patch and subsequently identified the center of the patch with the highest Shapley value as the optimal cropping center.

### A.2 PRETRAINED TEACHER MODEL

When generating soft labels, we utilized teacher models from the early stages of training. Specifically, for CIFAR-10 and CIFAR-100, the teacher models were pretrained for 10 epochs using a learn-

---

[1] https://captum.ai/api/shapley_value_sampling.html

Table 9: Comparison of Top-1 Accuracy (%) of InfoUtil vs. Coreset Selection Methods (ConvNet).

| Model | Dataset | IPC | Random | Herding | Forgetting | InfoUtil (Ours) |
|---|---|---|---|---|---|---|
| ConvNet | CIFAR-10 | 1 | $14.4 \pm 2.0$ | $21.5 \pm 1.2$ | $13.5 \pm 1.2$ | $\mathbf{28.5 \pm 1.4}$ |
| | | 10 | $26.0 \pm 1.2$ | $31.6 \pm 0.7$ | $23.3 \pm 1.0$ | $\mathbf{54.1 \pm 0.5}$ |
| | | 50 | $43.4 \pm 1.0$ | $40.4 \pm 0.6$ | $23.3 \pm 1.1$ | $\mathbf{69.8 \pm 0.1}$ |
| | CIFAR-100 | 1 | $4.2 \pm 0.3$ | $8.4 \pm 0.3$ | $4.5 \pm 0.2$ | $\mathbf{33.1 \pm 0.3}$ |
| | | 10 | $14.6 \pm 0.5$ | $17.3 \pm 0.3$ | $15.1 \pm 0.3$ | $\mathbf{50.5 \pm 0.3}$ |
| | | 50 | $30.0 \pm 0.4$ | $33.7 \pm 0.5$ | $30.5 \pm 0.3$ | $\mathbf{57.8 \pm 0.2}$ |
| | Tiny ImageNet | 1 | $1.4 \pm 0.1$ | $1.4 \pm 0.1$ | $1.6 \pm 0.1$ | $\mathbf{19.6 \pm 0.5}$ |
| | | 10 | $5.0 \pm 0.2$ | $5.0 \pm 0.2$ | $5.1 \pm 0.2$ | $\mathbf{40.2 \pm 0.3}$ |
| | | 50 | $15.0 \pm 0.4$ | $15.0 \pm 0.4$ | $15.0 \pm 0.3$ | $\mathbf{48.0 \pm 0.5}$ |

Table 10: Comparison of Top-1 Accuracy (%) of InfoUtil vs. Coreset Selection (ResNet-18).

| Model | Dataset | IPC | Random | Herding | K-Means | InfoUtil (Ours) |
|---|---|---|---|---|---|---|
| ResNet-18 | Tiny ImageNet | 10 | $7.5 \pm 0.1$ | $9.0 \pm 0.3$ | $8.9 \pm 0.2$ | $\mathbf{45.6 \pm 0.3}$ |
| | ImageNet-1K | 10 | $4.4 \pm 0.1$ | $5.8 \pm 0.1$ | $5.5 \pm 0.1$ | $\mathbf{44.2 \pm 0.4}$ |

ing rate of 0.001 on IPC = 1 and 10. Meanwhile, for other datasets (Tiny-ImageNet, ImageNette, ImageWoof, ImageNet-100, and ImageNet-1k), we trained the teacher models for 10 epochs using a learning rate of 0.01 on IPC = 1 and 10. For IPC = 50 scenarios, we employed fully converged teacher models across all datasets to ensure that soft labels generated could reflect the comprehensive and stable representations learned from the entire training dataset. Compared to teacher models from early training stages, fully converged models provide richer, more accurate semantic information, which significantly benefit the distillation process, especially when synthesizing a larger number of representative images.

## B  CORESET SELECTION COMPARISON AND INFORMATION DENSITY

We investigate the fundamental distinction between data synthesis (InfoUtil) and traditional coreset selection methods, which aim to construct a compact dataset by selecting unaltered real samples. While both approaches pursue dataset compression, InfoUtil's ability to synthesize highly informative, compressed knowledge yields a significant performance gap, especially under extreme data scarcity (IPC = 1 or 10).

### B.1  FUNDAMENTAL DISTINCTION AND EMPIRICAL ADVANTAGE

Traditional coreset selection methods (such as Random, Herding Welling (2009) and Forgetting (Toneva et al., 2018)) are constrained by the quality and content of the original training samples. InfoUtil overcomes this limitation by dynamically synthesizing samples that are optimized for knowledge transfer, extracting informative patches, and utilizing soft labels to condense teacher knowledge. As shown in Tab. 9 and Tab. 10, to empirically demonstrate this advantage, we compare InfoUtil against coreset selection baselines across CIFAR, Tiny-ImageNet, and ImageNet-1K.

### B.2  PERFORMANCE ON LARGE-SCALE DATASETS

As shown in Tab. 10, we further validate the results using a deeper architecture (ResNet-18) on challenging large-scale datasets, comparing against K-Means coreset selection.

The empirical results show a massive performance gap. For example, on ImageNet-1K (IPC = 10), InfoUtil achieves $44.2\%$, which is nearly 7.6 times higher than the best coreset method (Herding, $5.8\%$). This clearly illustrates that simply selecting real images is insufficient for training deep networks from scratch on such limited budgets. Coreset methods inherently suffer from background noise and reliance on hard labels. In contrast, InfoUtil's synthesis mechanism which incorporates attribution cropping (Informativeness) and soft labels, effectively condenses the necessary knowledge, making it far more efficient and powerful than standard subset selection.

Table 11: Accuracy (%) under the Controlled "Fully Converged Teacher" Setting on ImageWoof.

| Model | Method | IPC=1 | IPC=10 | IPC=50 |
|---|---|---|---|---|
| ConvNet | RDED | 18.5 | 40.6 | 61.5 |
| | InfoUtil (Ours) | **20.0** | **42.4** | **62.6** |
| ResNet-18 | RDED | 20.8 | 38.5 | 68.5 |
| | InfoUtil (Ours) | **21.4** | **43.6** | **69.2** |
| ResNet-101 | RDED | 19.6 | 31.3 | 59.1 |
| | InfoUtil (Ours) | **19.8** | **35.0** | **67.0** |

## C ANALYSIS OF SOFT LABELING STRATEGY ROBUSTNESS

To ensure a fair performance assessment, we rigorously isolate the contribution of our proposed InfoUtil from potential advantages conferred by the soft-labeling strategy employed by the teacher model. While previous distillation literature has explored utilizing "early-stage teacher" models for maximizing performance at low Images Per Class (IPC) settings, we demonstrate the intrinsic robustness of InfoUtil by unifying the teacher protocol.

### C.1 CONTROLLED EXPERIMENT WITH FULLY CONVERGED TEACHER

We conducted a controlled experiment on the ImageWoof dataset where both the baseline (RDED) and our InfoUtil method were strictly constrained to use the exact same Fully Converged Teacher model across all tested IPC settings (1, 10, and 50). This experimental setup eliminates any potential performance artifact stemming from differences in teacher model convergence stages, ensuring that measured gains are attributed solely to InfoUtil's data synthesis mechanism (Shapley-based informativeness and GradNorm utility).

The results across different student architectures (ConvNet, ResNet-18, and ResNet-101) are reported in Table 11. As evidenced by Table 11, our method consistently maintains a clear performance advantage over the RDED baseline under this strict controlled setting, with improvements observed across every student architecture and IPC configuration. The gains are particularly substantial in deeper architectures and higher compression rates (e.g., a $7.9\%$ margin for ResNet-101 at IPC $= 50$). This data strongly validates that the performance gains are not an artifact of the labeling strategy but are directly attributable to InfoUtil's core mechanism of selecting and synthesizing high-informativeness data.

## D PROOFS OF SHAPLEY VALUE AXIOMS

Building upon the game-theoretic formulation in Section 3, we now formally show that our feature attribution method—which maximizes informativeness via Shapley values—satisfies the four axiomatic properties of Shapley values. These properties ensure that the attributions assigned to input variables are theoretically sound and fair.

Consistent with our informativeness maximization framework, we define:

- Neural network as characteristic function: The deep neural network $f$ acts as the characteristic function in a cooperative game, mapping each coalition of features to a predictive score.
- Players: Each input variable $x^{(i)}$ ($i \in [d] := \{1, 2, \ldots, d\}$) is treated as a distinct player in the game.
- Coalitions: A binary mask $s \in \{0, 1\}^d$ represents a coalition of active features, with $s_i = 1$ indicating inclusion of $x^{(i)}$ and $s_i = 0$ indicating its exclusion.
- Reward: The informativeness score $f(s \circ x)$ is regarded as the reward contributed by the coalition $s$.

The Shapley value for variable $x^{(i)}$ is computed as:

$$\phi_f(x^{(i)}) = \frac{1}{d} \sum_{s:s_i=0} \binom{d-1}{\mathbf{1}^\top s} \left( f(x \circ (s + e_i)) - f(x \circ s) \right),$$

where $e_i \in \mathbb{R}^d$ denotes the vector with a one in the $i$-th position but zeros in the rest positions, and $s$ is a binary mask indicating active input variables.

## D.1 PROOF OF AXIOM 1(LINEARITY)

**Axiom 1 (Linearity)** *If two games can be merged into a new game, then the Shapley Values in the two original games can also be merged. Formally, if $f_{\mathrm{merged}} = f_1 + f_2$, then $\phi_{f_{\mathrm{merged}}}(x^{(i)}) = \phi_{f_1}(x^{(i)}) + \phi_{f_2}(x^{(i)}), \forall i \in [d]$.*

**Proof of Axiom 1:** For merged game $f_{\mathrm{merged}} = f_1 + f_2$, by definition we have $f_{\mathrm{merged}}(x \circ t) = f_1(x \circ t) + f_2(x \circ t)$ for any mask $t$. Substituting into the Shapley value formula:

$$
\begin{aligned}
\phi_{f_{\mathrm{merged}}}(x^{(i)}) &= \frac{1}{d} \sum_{s:s_i=0} \binom{d-1}{\mathbf{1}^\top s} \left( f_{\mathrm{merged}}(x \circ (s+e_i)) - f_{\mathrm{merged}}(x \circ s) \right) \\
&= \frac{1}{d} \sum_{s:s_i=0} \binom{d-1}{\mathbf{1}^\top s} \left[ (f_1(x \circ (s+e_i)) + f_2(x \circ (s+e_i))) - (f_1(x \circ s) + f_2(x \circ s)) \right] \\
&= \frac{1}{d} \sum_{s:s_i=0} \binom{d-1}{\mathbf{1}^\top s} (f_1(x \circ (s+e_i)) - f_1(x \circ s)) + \\
&\quad \frac{1}{d} \sum_{s:s_i=0} \binom{d-1}{\mathbf{1}^\top s} (f_2(x \circ (s+e_i)) - f_2(x \circ s)) \\
&= \phi_{f_1}(x^{(i)}) + \phi_{f_2}(x^{(i)}).
\end{aligned}
$$

Thus, if $f_{\mathrm{merged}} = f_1 + f_2$, then $\phi_{f_{\mathrm{merged}}}(x^{(i)}) = \phi_{f_1}(x^{(i)}) + \phi_{f_2}(x^{(i)}), \forall i \in [d]$.

## D.2 PROOF OF AXIOM 2(DUMMY)

**Axiom 2 (Dummy)** *A dummy player $i$ is a player that has no interactions with other players in the game $f$. Formally, if $\forall s : s_i = 0$, $f(x \circ (s + e_i)) = f(x \circ s) + f(x \circ e_i)$. Then, the dummy player's Shapley Value is computed as $f(x \circ e_i)$.*

**Proof of Axiom 2:** For a dummy player $i$ satisfying $\forall s : s_i = 0, f(x \circ (s+e_i)) = f(x \circ s) + f(x \circ e_i)$, substitute the condition into the Shapley value formula:

$$
\begin{aligned}
\phi_f(x^{(i)}) &= \frac{1}{d} \sum_{s:s_i=0} \binom{d-1}{\mathbf{1}^\top s} (f(x \circ (s+e_i)) - f(x \circ s)) \\
&= \frac{1}{d} \sum_{s:s_i=0} \binom{d-1}{\mathbf{1}^\top s} (f(x \circ e_i)) \\
&= f(x \circ e_i) \cdot \frac{1}{d} \sum_{s:s_i=0} \binom{d-1}{\mathbf{1}^\top s}.
\end{aligned}
$$

Note that the sum over all $s : s_i = 0$ (subsets of the remaining $d - 1$ variables) satisfies:

$$
\sum_{s:s_i=0} \binom{d-1}{\mathbf{1}^\top s} = \sum_{k=0}^{d-1} \binom{d-1}{k} = 2^{d-1} \cdot \frac{d}{d} = d,
$$

where we use the identity $\sum_{k=0}^{n} \binom{n}{k} = 2^n$ with $n = d - 1$. Thus:

$$
\phi_f(x^{(i)}) = f(x \circ e_i) \cdot \frac{1}{d} \cdot d = f(x \circ e_i).
$$

## D.3 PROOF OF AXIOM 3(SYMMETRY)

**Axiom 3 (Symmetry)** *If two players contribute equally in every case, then their Shapley values in the game $f$ will be equal. Formally, if $\forall s : s_i = s_j = 0$, $f(x \circ (s + e_i)) = f(x \circ (s + e_j))$, then $\phi_f(x^{(i)}) = \phi_f(x^{(j)})$.*

**Proof of Axiom 3:** For symmetric players $i$ and $j$ satisfying $\forall s : s_i = s_j = 0$, $f(x \circ (s + e_i)) = f(x \circ (s + e_j))$, consider their Shapley values:

$$\phi_f(x^{(i)}) = \frac{1}{d} \sum_{s:s_i=0} \binom{d-1}{\mathbf{1}^\top s} \left( f(x \circ (s + e_i)) - f(x \circ s) \right),$$

$$\phi_f(x^{(j)}) = \frac{1}{d} \sum_{s:s_j=0} \binom{d-1}{\mathbf{1}^\top s} \left( f(x \circ (s + e_j)) - f(x \circ s) \right).$$

Define a bijection between masks $s : s_i = 0$ and $t : t_j = 0$ via $t = s$ if $j \notin s$, and $t = (s \setminus \{j\}) \cup \{i\}$ if $j \in s$. By symmetry, $f(x \circ (s + e_i)) = f(x \circ (t + e_j))$ and $f(x \circ s) = f(x \circ t)$. Since $\mathbf{1}^\top s = \mathbf{1}^\top t$, the binomial coefficients are equal. Thus:

$$\phi_f(x^{(i)}) = \frac{1}{d} \sum_{t:t_j=0} \binom{d-1}{\mathbf{1}^\top t} \left( f(x \circ (t + e_j)) - f(x \circ t) \right) = \phi_f(x^{(j)}).$$

### D.4 Proof of Axiom 4(Efficiency)

**Axiom 4 (Efficiency)** *The total reward of the game $f$ is equal to the sum of the Shapley values of all players. Formally, $f(x) - f(\mathbf{0}) = \sum_{i \in [d]} \phi_f(x^{(i)})$.*

**Proof of Axiom 4:** Summing Shapley values over all players:

$$\sum_{i \in [d]} \phi_f(x^{(i)}) = \sum_{i \in [d]} \frac{1}{d} \sum_{s:s_i=0} \binom{d-1}{\mathbf{1}^\top s} \left( f(x \circ (s + e_i)) - f(x \circ s) \right)$$

$$= \frac{1}{d} \sum_{s \subseteq [d]} \sum_{i \notin s} \binom{d-1}{\mathbf{1}^\top s} \left( f(x \circ (s + e_i)) - f(x \circ s) \right).$$

For a fixed mask $s$ with $\mathbf{1}^\top s = k$, there are $d - k$ players not in $s$. The inner sum becomes:

$$\sum_{i \notin s} \left( f(x \circ (s + e_i)) - f(x \circ s) \right) = \sum_{i \notin s} f(x \circ (s + e_i)) - (d - k) f(x \circ s).$$

Summing over all $s$ and telescoping the series, all intermediate terms cancel, leaving:

$$\sum_{i \in [d]} \phi_f(x^{(i)}) = f(x) - f(\mathbf{0}).$$

## E Proofs of Theorems

This appendix presents the full derivation to formally establish Theorem 1, complementing the partial analysis in the main text.

Recall the definition of utility:

**Theorem 1: Utility is bounded by Gradient Norm.** Let the utility function $\mathcal{U}$ be defined as in Definition 3. Then there exists a constant $c > 0$ such that

$$\mathcal{U}(x_i, y_i; f_{\theta^{(t)}}) \le c \|\nabla_{\theta^{(t)}} \ell_t(f_{\theta^{(t)}}(x_i), y_i)\|.$$

Using the chain rule for gradient flow, we have

$$\dot{\ell}_t(f_{\theta^{(t)}}(x_j), y_j; \mathcal{B}) = \nabla_{\theta^{(t)}} \ell_t(f_{\theta^{(t)}}(x_j), y_j) \cdot \left. \frac{\partial \theta^{(t)}}{\partial t} \right|_{\mathcal{B}},$$

and similarly for $\mathcal{B}_{\neg i}$. Thus, the change in gradient flow is

$$\left| \dot{\ell}_t(f_{\theta^{(t)}}(x_j), y_j; \mathcal{B}) - \dot{\ell}_t(f_{\theta^{(t)}}(x_j), y_j; \mathcal{B}_{\neg i}) \right| = \left| \nabla_{\theta^{(t)}} \ell_t(f_{\theta^{(t)}}(x_j), y_j) \cdot \left( \left. \frac{\partial \theta^{(t)}}{\partial t} \right|_{\mathcal{B}} - \left. \frac{\partial \theta^{(t)}}{\partial t} \right|_{\mathcal{B}_{\neg i}} \right) \right|.$$

Under SGD with learning rate $\eta$, the update step is

$$\frac{\partial \theta^{(t)}}{\partial t}\bigg|_{\mathcal{B}} = -\eta \sum_{(x,y)\in\mathcal{B}} \nabla_{\theta^{(t)}} \ell_t(f_{\theta^{(t)}}(x), y).$$

Removing $(x_i, y_i)$ gives

$$\frac{\partial \theta^{(t)}}{\partial t}\bigg|_{\mathcal{B}_{\neg i}} = -\eta \sum_{(x,y)\in\mathcal{B}_{\neg i}} \nabla_{\theta^{(t)}} \ell_t(f_{\theta^{(t)}}(x), y).$$

Taking the difference,

$$\frac{\partial \theta^{(t)}}{\partial t}\bigg|_{\mathcal{B}} - \frac{\partial \theta^{(t)}}{\partial t}\bigg|_{\mathcal{B}_{\neg i}} = -\eta \nabla_{\theta^{(t)}} \ell_t(f_{\theta^{(t)}}(x_i), y_i).$$

Substituting this into the gradient flow change gives

$$\left| \dot{\ell}_t(f_{\theta^{(t)}}(x_j), y_j; \mathcal{B}) - \dot{\ell}_t(f_{\theta^{(t)}}(x_j), y_j; \mathcal{B}_{\neg i}) \right|$$
$$= \eta \left| \nabla_{\theta^{(t)}} \ell_t(f_{\theta^{(t)}}(x_j), y_j) \cdot \nabla_{\theta^{(t)}} \ell_t(f_{\theta^{(t)}}(x_i), y_i) \right|$$
$$\leq \eta \| \nabla_{\theta^{(t)}} \ell_t(f_{\theta^{(t)}}(x_j), y_j) \| \cdot \| \nabla_{\theta^{(t)}} \ell_t(f_{\theta^{(t)}}(x_i), y_i) \|$$

where the last step follows from the Cauchy–Schwarz inequality. Let

$$c = \eta \max_{(x_j, y_j)\in\mathcal{D}} \| \nabla_{\theta^{(t)}} \ell_t(f_{\theta^{(t)}}(x_j), y_j) \|$$

be a constant independent of $(x_i, y_i)$. Taking the maximum over $(x_j, y_j) \in \mathcal{D}$, we obtain

$$\mathcal{U}(x_i, y_i; f_{\theta^{(t)}}) \leq c \| \nabla_{\theta^{(t)}} \ell_t(f_{\theta^{(t)}}(x_i), y_i) \|.$$

Note that $c = \eta \max_{(x_j, y_j)\in\mathcal{D}} \| \nabla_{\theta^{(t)}} \ell_t(f_{\theta^{(t)}}(x_j), y_j) \|$ satisfies the following properties:

1. It is independent of the current measured data $(x_i, y_i)$, ensuring that the bound in Theorem 1 holds uniformly for all training examples.
2. It only assumes that the gradient norm $\| \nabla_{\theta^{(t)}} \ell_t(f_{\theta^{(t)}}(x_j), y_j) \|$ has an upper bound, which is a reasonable assumption for any successfully converged model.
3. Since the learning rate $\eta$ can be chosen to be small in practice, the value of $c$ remains controlled and does not become excessively large.

## F  ADDITIONAL VISUALIZATIONS OF SYNTHETIC DATA

Compared to optimization-based SRe2L, InfoUtil creates more realistic images by preserving details and color consistency. Compared to optimization-free methods like RDED, InfoUtil stands out with its enhanced interpretability and structured framework, emphasizing key semantic details while reducing focus on irrelevant areas. We present further visual comparisons of synthetic ImageNet-1K images generated by SRe2L, RDED, and InfoUtil at IPC = 10. Specifically, Figures 6, 7, 8 and 9 illustrate results across three representative classes, clearly highlighting the superior visual quality attained by InfoUtil. As intuitive evidence, Figure 5 shows condensed images for ImageNet-1K's *indigo bunting* category, including results from Original, Herding, SRe2L, RDED, and InfoUtil (Ours). InfoUtil focuses on the most discriminative object parts, yielding more informative results. Additional visualizations are provided in the supplements due to space constraints.

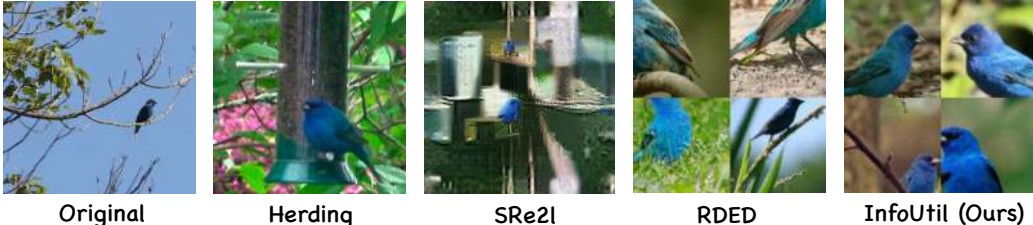

Figure 5: Visualization of condensed images for the indigo bunting category on ImageNet-1K.

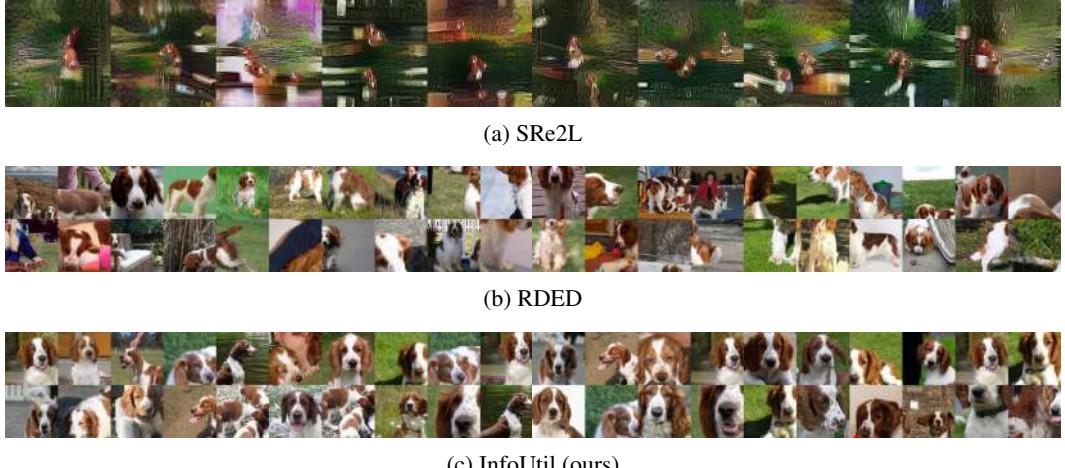

(a) SRe2L

(b) RDED

(c) InfoUtil (ours)

Figure 6: We visualized synthesized images generated by SOTA methods and InfoUtil on ImageNet-1K. These images are distilled from the "Welsh Springer Spaniel" category.

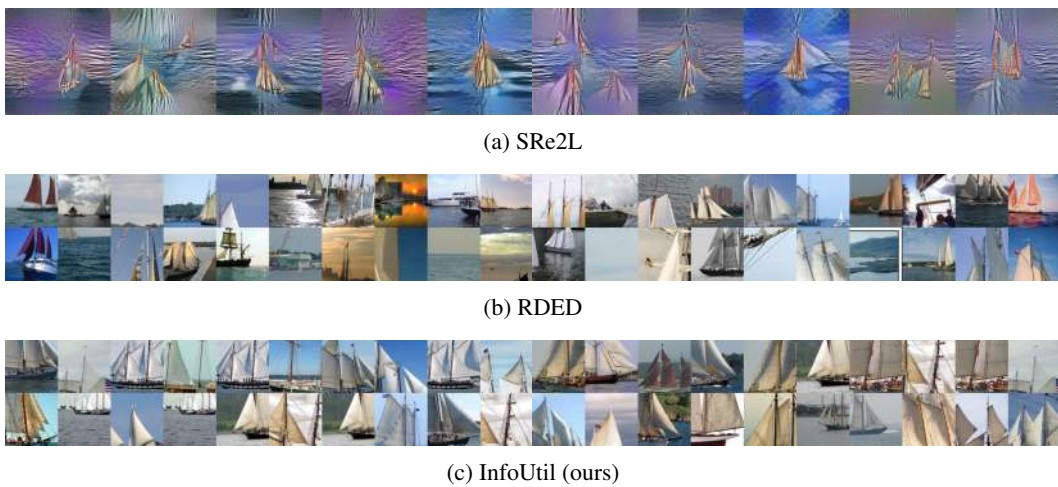

Figure 7: We visualized synthesized images generated by SOTA methods and InfoUtil on ImageNet-1K. These images are distilled from the "schooner" category.

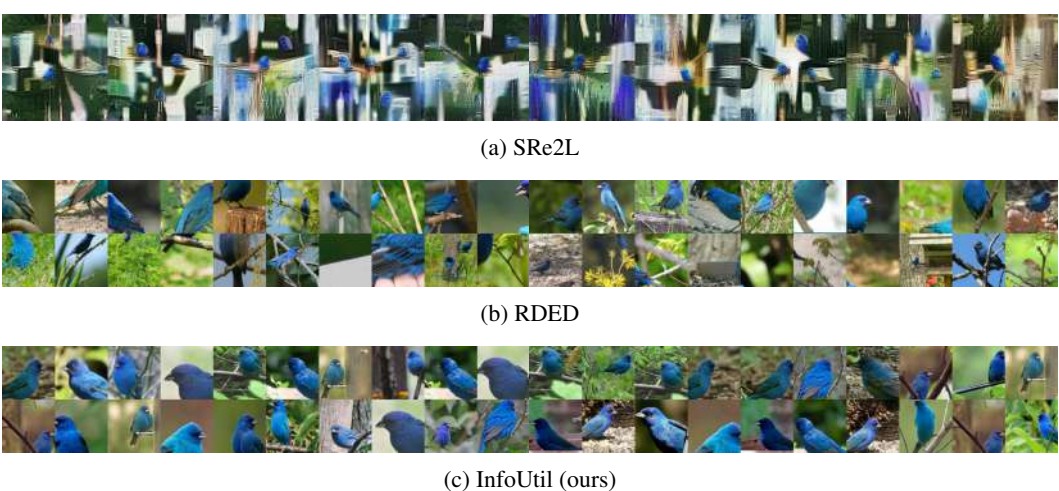

Figure 8: We visualized synthesized images generated by SOTA methods and InfoUtil on ImageNet-1K. These images are distilled from the "indigo bunting" category.

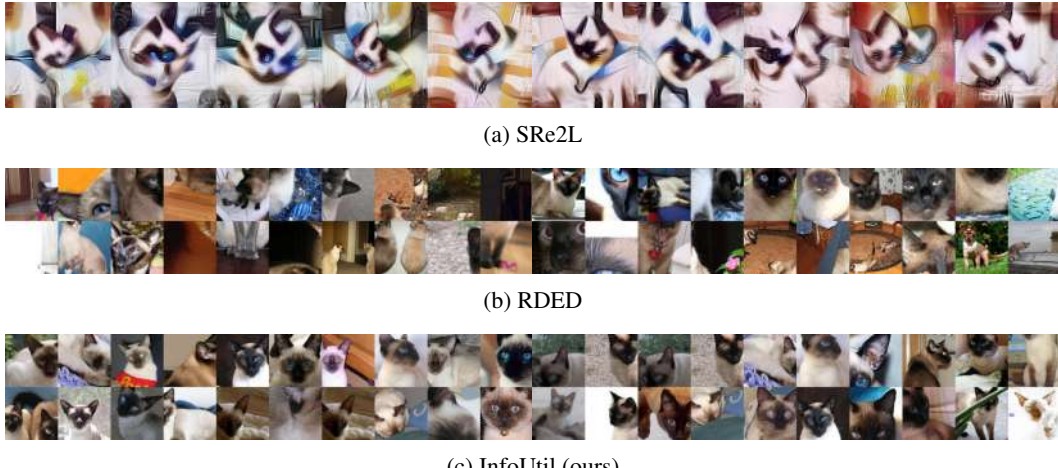

Figure 9: We visualized synthesized images generated by SOTA methods and InfoUtil on ImageNet-1K. These images are distilled from the "Siamese cat" category.

