# OpenReview forum: "Grounding and Enhancing Informativeness and Utility in Dataset Distillation"
_ICLR.cc/2026/Conference — ICLR 2026 Poster_

### Official Review · Reviewer_CGAE · 2025-10-24

**Soundness:** 2
**Presentation:** 3
**Contribution:** 2
**Rating:** 4
**Confidence:** 4

**Summary:**

This paper proposes InfoUtil including two key concepts: Informativeness (how much useful information a sample carries) and Utility (how essential a sample is for training). InfoUtil maximizes these objectives via (1) Shapley Value-based informativeness attribution and (2) Gradient Norm-based utility selection. Experiments on ImageNet-1K with ResNet-18 show a 6.1% improvement over prior SOTA.

**Strengths:**

The presentation is clear and well-structured.

The theoretical analysis  is reasonably justified.

**Weaknesses:**

I noticed that the authors carefully adopt different levels of teacher models to generate soft labels for different IPC settings. However, since this labeling strategy is directly borrowed from prior works rather than being a core contribution of this submission, it should be consistently applied to all comparison methods to ensure fairness, at least to RDED. Otherwise, part of the reported performance gain may be attributed to better soft labeling rather than the proposed method itself.

The performance on ConvNet with CIFAR is significantly weaker than that of recent competitors. In addition, several of the compared distillation-based baselines are outdated. Pls consider some newer and stronger baselines such as
- Elucidating the Design Space of Dataset Condensation
- Dataset Distillation via the Wasserstein Metric
- Breaking Class Barriers: Efficient Dataset Distillation via Inter-Class Feature Compensator.
- Heavy Labels Out! Dataset Distillation with Label Space Lightening

The current ablation is also insufficient. The contributions of GradNorm Scoring and Attribution Cropping should be clearly isolated from the carefully designed soft labeling strategy in order to demonstrate their real effectiveness. Furthermore, in large IPC settings, it is unclear why SRe²L and RDED are missing, especially given that both are already open-sourced and widely used.

Lines 435–436 in Table 6 report identical settings (both GradNorm Scoring and Attribution Cropping), yet the numbers differ. Please verify and correct.

Finally, there are typos, e.g., “scriptsizeImageNet.” in line 364.

**Questions:**

pls see the weaknesses.

---

> ### Author Response · Authors · 2025-11-21
> **(Part 1) Reponse to Reviewer CGAE**
>
> ### Q1: About soft-label generation.
>
> Thanks for your this insightful comment. We agree that isolating the contribution of our proposed method from the soft-labeling strategy is crucial for a fair comparsion.
>
> **1. Clarification on the Experimental Protocol:** While we originally adopted the "early-stage teacher" strategy for low IPC settings following insights from recent literature \[1,2\] to maximize performance, we acknowledge the reviewer's concern that this might obscure the intrinsic gains of InfoUtil.
>
> **2. Controlled Experiment (Same Fully Converged Teacher):** To directly address this concern and demonstrate the robustness of our method, we conducted a new controlled experiment on the ImageWoof dataset. In this setup, we strictly unified the soft-labeling strategy: both the baseline (RDED) and our method (InfoUtil) used the exact same Fully Converged Teacher model across all IPC settings (1, 10, and 50), eliminating any advantage from teacher tuning.
>
> The results are reported in the table below:
> | Model | Method | IPC=1 | IPC=10 | IPC=50 |
> | :---: | :---: | :---: | :---: | :---: |
> | ConvNet | RDED | 18.5 | 40.6 | 61.5 |
> | | InfoUtil | 20 | 42.4 | 62.6 |
> | Resnet-18 | RDED | 20.8 | 38.5 | 68.5 |
> | | InfoUtil | 21.4 | 43.6 | 69.2 |
> | Resnet-101 | RDED | 19.6 | 31.3 | 59.1 |
> | | InfoUtil | 19.8 | 35 | 67 |
>
> As evidenced by the table, our method maintains a clear performance advantage over RDED under the strict "Fully Converged" setting, with improvements observed across every architecture and IPC configuration. The gains are particularly substantial in certain cases, such as the 5.1% jump for ResNet-18 at IPC=10 and the notable 7.9% margin for ResNet-101 at IPC=50. This data strongly validates that the performance gains are not an artifact of the labeling strategy but are directly attributable to InfoUtil's core mechanisms of selecting data based on Shapley-based informativeness and GradNorm-assessed utility.
>
> We have added these additional results and discussions to the appendix of the revised manuscript.
>
> \[1\] Qin T, Deng Z, Alvarez-Melis D. A label is worth a thousand images in dataset distillation[J]. Advances in Neural Information Processing Systems, 2024, 37: 131946-131971.
>
> \[2\] Sucholutsky I, Schonlau M. Soft-label dataset distillation and text dataset distillation[C]//2021 International Joint Conference on Neural Networks (IJCNN). IEEE, 2021: 1-8.
>
> We have also added related results in Appendix N(Table 19) in the revised manuscript.

---

> ### Author Response · Authors · 2025-11-21
> **(Part 2) Reponse to Reviewer CGAE**
>
> ### Q2: Ask for more baseline comparison.
>
> We thank the reviewer for pointing out these competitive and recent baselines. We agree that comparing against these methods is essential to demonstrate the effectiveness of InfoUtil.
>
> **1. Comprehensive Comparison:**
> Following your suggestion, we have conducted extensive additional experiments to compare InfoUtil against:
>
> *    **WMDD:** Dataset Distillation via the Wasserstein Metric \[3\].
> *    **INFER:** Breaking class barriers: Efficient dataset distillation via inter-class feature compensator \[4\].
> *    **HeLLo:** Heavy Labels Out! Dataset Distillation with Label Space Lightening \[5\].
>
> | Dataset (ResNet-18) | IPC | SRe2L | WMDD | RDED | InfoUtil |
> | :---: | :---: | :---: | :---: | :---: | :---: |
> | ImageNette | 1 | 19.1 $\pm$ 1.1 | 40.2 $\pm$ 0.6 | 35.8 $\pm$ 1.0 | $\mathbf{43.8 \pm 0.7}$ |
> | | 10 | 29.4 $\pm$ 3.0 | 64.8 $\pm$ 0.4 | 61.4 $\pm$ 0.4 | $\mathbf{68.6 \pm 0.6}$ |
> | | 50 | 40.9 $\pm$ 0.3 | 83.5 $\pm$ 0.3 | 80.4 $\pm$ 0.4 | $\mathbf{86.2 \pm 0.6}$ |
> | Tiny-ImageNet | 1 | 2.6 $\pm$ 0.1 | 7.6 $\pm$ 0.2 | 9.7 $\pm$ 0.4 | $\mathbf{17.0 \pm 1.3}$ |
> | | 10 | 16.1 $\pm$ 0.2 | 41.8 $\pm$ 0.1 | 41.9 $\pm$ 0.2 | $\mathbf{45.6 \pm 0.3}$ |
> | | 50 | 41.1 $\pm$ 0.4 | $\mathbf{59.4 \pm 0.5}$ | 58.2 $\pm$ 0.1 | 58.5 $\pm$ 0.3 |
> | ImageNet-1K | 1 | 0.1 $\pm$ 0.1 | 3.2 $\pm$ 0.3 | 6.6 $\pm$ 0.2 | $\mathbf{12.7 \pm 0.7}$ |
> | | 10 | 21.3 $\pm$ 0.6 | 38.2 $\pm$ 0.2 | 42.0 $\pm$ 0.1 | $\mathbf{44.2 \pm 0.4}$ |
> | | 50 | 46.8 $\pm$ 0.2 | 57.6 $\pm$ 0.5 | 56.5 $\pm$ 0.1 | $\mathbf{58.0 \pm 0.3}$ |
>
>
> | Dataset (ResNet-18) | IPC | SRe2L | INFER | RDED | InfoUtil |
> | :---: | :---: | :---: | :---: | :---: | :---: |
> | Cifar10 | 10 | 29.3 $\pm$ 0.5 | 30.7 $\pm$ 0.3 | 37.1 $\pm$ 0.3 | $\mathbf{53.8 \pm 0.1}$ |
> | | 50 | 45.0 $\pm$ 0.7 | 60.7 $\pm$ 0.9 | 62.1 $\pm$ 0.1 | $\mathbf{71.0 \pm 1.4}$ |
> | Tiny-ImageNet | 10 | 16.1 $\pm$ 0.2 | 41.0 $\pm$ 0.4 | 41.9 $\pm$ 0.2 | $\mathbf{45.6 \pm 0.3}$ |
> | | 50 | 41.1 $\pm$ 0.4 | 54.6 $\pm$ 0.4 | 58.2 $\pm$ 0.1 | $\mathbf{58.5 \pm 0.3}$ |
> | ImageNet-1K | 50 | 46.8 $\pm$ 0.2 | 54.3 $\pm$ 0.6 | 56.5 $\pm$ 0.1 | $\mathbf{58.0 \pm 0.3}$ |
>
>
> | Dataset (ResNet-18) | IPC | SRe2L | RDED | HeLlo | InfoUtil |
> | :---: | :---: | :---: | :---: | :---: | :---: |
> | ImageNet-100 | 1 | 3.0 $\pm$ 0.3 | 8.1 $\pm$ 0.3 | 12.5 $\pm$ 0.2 | $\mathbf{15.7 \pm 0.2}$ |
> | | 10 | 9.5 $\pm$ 0.4 | 36.0 $\pm$ 0.3 | 48.9 $\pm$ 0.1 | $\mathbf{50.5 \pm 0.4}$ |
> | | 50 | 27.0 $\pm$ 0.4 | 61.6 $\pm$ 0.1 | $\mathbf{69.4 \pm 0.1}$ | 68.3 $\pm$ 0.4 |
> | ImageNet-1K | 1 | 0.1 $\pm$ 0.1 | 6.6 $\pm$ 0.2 | $\mathbf{12.9 \pm 0.3}$ | 12.7 $\pm$ 0.7 |
> | | 10 | 21.3 $\pm$ 0.6 | 42.0 $\pm$ 0.1 | 43.7 $\pm$ 0.1 | $\mathbf{44.2 \pm 0.4}$ |
> | | 50 | 46.8 $\pm$ 0.2 | 56.5 $\pm$ 0.1 | 52.2 $\pm$ 0.1 | $\mathbf{58.0 \pm 0.3}$ |
>
>
>
> **2. Performance Analysis:**
> The consolidated results demonstrate that InfoUtil achieves state-of-the-art performance in the vast majority of settings:
>
> *    **Strong Performance on CIFAR-10:** Addressing the reviewer's concern about weak performance on ConvNet, InfoUtil achieves **71.0%** (IPC=50) on CIFAR-10, significantly outperforming INFER (60.7%) by **10.3%** and SRe2L (45.0%) by **26.0%**.
> *    **Dominance in Low-Data Regimes:** InfoUtil excels in challenging low-IPC settings. On **Tiny-ImageNet (IPC=1)**, our method achieves **17.0%**, surpassing WMDD (7.6%) by a remarkable margin of **+9.4%**. Similarly, on **ImageNet-100 (IPC=1)**, we outperform HeLLo (15.7% vs. 12.5%).
> *    **Scalability on ImageNet:** On the large-scale ImageNet-1K, InfoUtil maintains its lead at IPC=10 (**44.2%**), outperforming WMDD (38.2%) and HeLLo (43.7%).
>
> We observe that method **HeLLo** performs slightly better on ImageNet-100 at IPC=50 (69.4% vs. 68.3%), and **WMDD** is marginally higher on Tiny-ImageNet at IPC=50 (59.4% vs. 58.5%). These methods utilize specialized optimization for label spaces or metrics which benefit high-budget regimes. However, InfoUtil provides a better trade-off, offering superior performance in the more difficult low-IPC regimes and maintaining top-tier performance across the board.
>
> The inclusion of these baselines confirms that InfoUtil establishes a new SOTA across most benchmarks.

---

> ### Author Response · Authors · 2025-11-21
> **(Part 3) Reponse to Reviewer CGAE**
>
> **3. About EDC \[6\]:**
> We acknowledge that EDC is a strong baseline representing the state-of-the-art in Training-Based (TB) methods. However, we would like to clarify the distinct advantages of our method compared to EDC:
>
> **a. Efficiency vs. Optimization Cost:** There is a fundamental trade-off between performance and synthesis efficiency.
> *    EDC is a Training-Based method that relies on complex bi-level optimization and extensive computation, requiring significant GPU hours to synthesize data.
> *    InfoUtil, in contrast, is designed for efficiency. It rapidly synthesizes data by extracting informative patches and selecting utility-optimized samples. This allows InfoUtil to generate high-quality datasets at a fraction of the computational cost required by optimization-based methods like EDC, making it far more practical for large-scale applications.
>
> **b. Superior Data Quality Validated via Initialization:**
> To further prove that the data synthesized by InfoUtil contains higher-quality information than previous methods (e.g., RDED), we conducted a validation experiment using **InfoUtil as the initialization** for EDC.
> The logic is straightforward: if InfoUtil provides a better starting point for the optimization process than RDED, it proves that InfoUtil's condensed patterns are intrinsically more valuable.
> We conducted experiments on **ImageNet-1K (ResNet-18)** with three settings:
> *    **Standard EDC:** Standard random/clustering initialization.
> *    **EDC + RDED Init:** Initialized with data from RDED.
> *    **EDC + InfoUtil Init:** Initialized with data from InfoUtil.
>
> The results provide strong evidence of InfoUtil's superior data quality:
> | Method | Initialization | IPC=1 | IPC=10 | IPC=50 |
> | :--- | :--- | :---: | :---: | :---: |
> | RDED [Sun et al.] | - | 6.6  | 42.0  | 56.5  |
> | EDC [Zhou et al.] | Standard | 12.8  | 48.6  | 58.0  |
> | EDC | + RDED Init | 12.9 | 48.8 | 58.2 |
> | **EDC** | **+ InfoUtil Init (Ours)** | **13.0** | **49.5** | **58.7** |
>
> \[3\] Liu H, Li Y, Xing T, et al. Dataset distillation via the wasserstein metric[C]//Proceedings of the IEEE/CVF International Conference on Computer Vision. 2025: 1205-1215.
>
> \[4\]Zhang X, Du J, Liu P, et al. Breaking class barriers: Efficient dataset distillation via inter-class feature compensator[J]. arXiv preprint arXiv:2408.06927, 2024.
>
> \[5\] Yu R, Liu S, Chen Z, et al. Heavy labels out! dataset distillation with label space lightening[C]//Proceedings of the IEEE/CVF International Conference on Computer Vision. 2025: 5017-5026.
>
> \[6\] Shao S, Zhou Z, Chen H, et al. Elucidating the design space of dataset condensation[J]. Advances in Neural Information Processing Systems, 2024, 37: 99161-99201.

---

> ### Author Response · Authors · 2025-11-21
> **(Part 4) Reponse to Reviewer CGAE**
>
> ### Q3: About ablation study on soft labels.
>
> We thank the reviewer for these constructive suggestions. We have addressed the concern regarding the isolation of contributions and completed the large IPC comparisons.
>
> **Isolating Core Contributions**. To clearly isolate the effectiveness of our method, we refer to the controlled experiment presented in Response Q1. In that experiment on ImageWoof, we strictly unified the soft-labeling strategy (using the same Fully Converged Teacher for both RDED and InfoUtil). The comparative results are summarized in the table below:
>
> | Model | Method | IPC=1 | IPC=10 | IPC=50 |
> | :---: | :---: | :---: | :---: | :---: |
> | ConvNet | RDED | 18.5 | 40.6 | 61.5 |
> | | InfoUtil | 20 | 42.4 | 62.6 |
> | Resnet-18 | RDED | 20.8 | 38.5 | 68.5 |
> | | InfoUtil | 21.4 | 43.6 | 69.2 |
> | Resnet-101 | RDED | 19.6 | 31.3 | 59.1 |
> | | InfoUtil | 19.8 | 35 | 67 |
>
> As observed, InfoUtil still consistently outperforms RDED even with the unified strategy. This confirms that the performance improvement is indeed driven by our proposed Informativeness (Attribution Cropping) and Utility (GradNorm Scoring) modules, rather than the labeling strategy.
>
> ### Q4: Ask for more results on large IPC settings.
>
> Thanks for your suggestion, we have conducted additional experiments on Tiny-ImageNet and ImageNet-1K under large IPC settings (up to 200), including the previously missing baselines SRe2L and RDED.
>
> The complete results are presented in the table below:
>
> | Dataset (ResNet-18) | IPC | SRe2L | RDED | InfoUtil |
> | :---: | :---: | :---: | :---: | :---: |
> | Tiny ImageNet | 50 | 41.1 $\pm$ 0.4 | 58.2 $\pm$ 0.1 | $\mathbf{58.5 \pm 0.3}$ |
> | | 100 | 49.7 $\pm$ 0.3 | 59.9 $\pm$ 0.4 | $\mathbf{60.6 \pm 0.5}$ |
> | | 200 | 51.2 $\pm$ 0.6 | 61.5 $\pm$ 0.3 | $\mathbf{62.0 \pm 0.3}$ |
> | ImageNet-1K | 10 | 21.3 $\pm$ 0.6 | 42.0 $\pm$ 0.1 | $\mathbf{43.5 \pm 0.4}$ |
> | | 50 | 46.8 $\pm$ 0.2 | 56.5 $\pm$ 0.1 | $\mathbf{57.6 \pm 0.3}$ |
> | | 100 | 52.8 $\pm$ 0.3 | 58.2 $\pm$ 0.6 | $\mathbf{58.8 \pm 0.4}$ |
> | | 200 | 57.0 $\pm$ 0.4 | 62.5 $\pm$ 0.8 | $\mathbf{63.4 \pm 0.3}$ |
>
> ### Q5: About table inconsistency.
>
> We apologize for this clerical error and sincerely thank the reviewer for the meticulous check.
>
> The intended setting for the second row is "GradNorm Scoring with Random Cropping". Therefore, the "Attribution Cropping" column for this row should be unmarked (×).
>
> We have corrected Table 6 in the revised manuscript. The corrected version is shown below:
>
> | GradNorm Scoring | Attribution Cropping | ImageWoof (IPC=1) | ImageWoof (IPC=50) | ImageNette (IPC=50) | ImageNet-1K (IPC=10) |
> | :---: | :---: | :---: | :---: | :---: | :---: |
> | ✗ | ✓ | 38.5 | 68.5 | 80.4 | 42.0 |
> | ✓ | ✗ | 43.6 | 68.8 | 85.0 | 43.5 |
> | ✓ | ✓ | 45.2 | 69.6 | 86.2 | 44.2 |
>
> Based on the corrected settings, the ablation results clearly validate the contribution of each component:
>
> *    **Effect of Attribution:** Comparing Row 2 (GradNorm only, 43.6% on ImageWoof IPC=1) with Row 3 (Full Method, 45.2%), the addition of Attribution Cropping yields a clear performance gain (+1.6%).
> *    **Effect of GradNorm:** Similarly, comparing Row 1 (Attribution only, 42.0% on ImageNet-1K) with Row 3 (Full Method, 44.2%), the inclusion of GradNorm Scoring provides significant improvements (+2.2%).
>
>
> ### Q6: About minor typos.
> We sincerely thank the reviewer for the careful reading and for pointing out this detail.
>
> We have corrected the typo "scriptsizeImageNet." in the revised manuscript. Additionally, we have thoroughly proofread the entire paper to fix other potential typos and grammatical errors to ensure the highest quality of presentation.

---

> ### Author Response · Authors · 2025-11-26
>
> Dear Reviewer CGAE,
>
> Thank you once again for your valuable comments on our submission. As the discussion phase is approaching its end, we would like to kindly confirm whether we have sufficiently addressed all of your concerns (or at least part of them). Should there be any remaining questions or areas requiring further clarification, please do not hesitate to let us know. If you are satisfied with our responses, we would greatly appreciate your consideration in adjusting the evaluation scores accordingly.
>
> We sincerely look forward to your feedback.

---

> ### Comment · Reviewer_CGAE · 2025-11-26
>
> Thank you for the rebuttal. The response addresses my concerns, and I am happy to raise my score. Please carefully check the writing and ensure that the rebuttal experiments are properly included.

---

> > ### Author Response · Authors · 2025-11-26
> >
> > Thanks for your recognition of the value of this work. We sincerely appreciate your recognition of our work and your support in raising the score.

---

### Official Review · Reviewer_987g · 2025-10-25

**Soundness:** 4
**Presentation:** 3
**Contribution:** 4
**Rating:** 8
**Confidence:** 3

**Summary:**

The paper introduces InfoUtil, a theoretically grounded framework for dataset distillation that aims to generate compact yet highly representative datasets. The method operates in two key stages. In the first stage, it employs the Shapley value, a game-theoretic attribution method, to identify the most informative image patches, ensuring that each synthetic sample captures the essential visual and semantic information from the original data. In the second stage, InfoUtil selects samples based on their gradient norms, which serves as a principled measure of utility by estimating the influence of each sample on model training dynamics. This dual optimization of informativeness and utility enables the distilled dataset to retain both rich content and strong learning potential. Extensive experiments across multiple benchmark datasets and architectures demonstrate that InfoUtil not only achieves superior performance but also provides interpretability and computational efficiency compared to existing methods, making it a robust and scalable approach to dataset distillation.

**Strengths:**

The strength of this work lies in its solid theoretical foundation and practical effectiveness. Unlike prior heuristic or empirical approaches, InfoUtil unifies the concepts of informativeness and utility within a principled mathematical framework, providing interpretability and transparency to dataset distillation. Its combination of game-theoretic Shapley value attribution and gradient norm-based sample selection ensures that the distilled data are both highly informative and influential for model training. Furthermore, InfoUtil achieves state-of-the-art performance across multiple benchmarks while being significantly more computationally efficient, requiring far less memory and training time than previous methods. This balance of theoretical rigor, interpretability, and scalability makes InfoUtil a robust and impactful contribution to the field of dataset distillation.

**Weaknesses:**

Despite its strong theoretical grounding and empirical performance, the paper contains a few minor weaknesses. There are some typo errors.

1. In lines 89–90, where the phrase "we reconsider the knowledge distillation-based dataset distillation process by introducing Principled Dataset Distillation (Definition 4)" should read "Optimal Dataset Distillation" to maintain consistency with the terminology used throughout the paper.
2. Similarly, in line 323, the word "nclude" in “Baseline nclude trajectory-matching" should be corrected to "include."

**Questions:**

A writing suggestion. In line 181-184, the authors should clarify which components are functions of \tilde{D}. Explicitly specifying the elements that depend on the distilled dataset would improve the mathematical clarity of the formulation and help readers better understand how \tilde{D} influences the optimization process.

---

> ### Author Response · Authors · 2025-11-21
> **Reponse to Reviewer 987g**
>
> ### Q1: About terminology consistency.
>
> Thanks for your suggestion. We have followed your advice to revise the terminology to ensure consistency.
> We have corrected the phrase in lines 89–90 to "Optimal Dataset Distillation" to ensure consistency with Definition 4 and the terminology used throughout the paper.
>
> ### Q2: About typos.
>
> Thanks for your advice. We have followed your advice to correct the typos.
> We have corrected the typo in line 323, changing "nclude" to "include".
>
> ### Q3: Writing suggestion.
>
> We appreciate this valuable suggestion to improve the mathematical clarity of our formulation.
>
> **1. Clarification of Dependency:** In Equation (1) and Definition 4, the distilled dataset $\tilde{D} = \{(x_j, y_j)\}_{j=1}^m$ is the decision variable of the optimization problem. Specifically:
>
> *   **In Equation (1):** The dependency lies in the model parameters $\theta_{\tilde{D}}$. The parameters $\theta_{\tilde{D}}$ are obtained by training the network $f$ on the distilled dataset $\tilde{D}$. Thus, the loss function $\mathcal{L}(f_{\theta_{\tilde{D}}}(x), y)$ is directly a function of $\tilde{D}$.
> *   **In Definition 4 (Optimal Dataset Distillation):** The objective is to select a subset $\tilde{D} \subseteq D'$ such that the total utility is maximized. Here, $\tilde{D}$ determines the domain of the summation. The optimization seeks to find the specific set of indices (samples) that maximizes the aggregated utility score.
>
> **2. Revision in Manuscript:**
>
> Following your advice, we have revised the text around lines 181-184 to explicitly specify these dependencies. The revised text reads:
>
> > "... where $\theta_{\tilde{D}}$ denotes the parameters of the model trained on the synthetic dataset $\tilde{D}$. Consequently, the term $\ell(f_{\theta_{\tilde{D}}}(x), y)$ depends on $\tilde{D}$ through the optimization trajectory of $\theta$..."
>
> We believe this clarification helps readers better understand how the selection of $\tilde{D}$ directly influences the final optimization objective.

---

### Official Review · Reviewer_tEAx · 2025-10-25

**Soundness:** 2
**Presentation:** 3
**Contribution:** 2
**Rating:** 6
**Confidence:** 4

**Summary:**

This paper revisits knowledge-distillation-based dataset distillation and introduces a principled framework named InfoUtil, which jointly optimizes Informativeness and Utility in the synthesis of compact datasets. The authors formally define informativeness at the patch level using a Shapley-value–based game-theoretic attribution, and utility at the sample level using gradient flow. The proposed two-step pipeline first selects the most informative image regions and then retains high-utility samples via gradient-norm scoring. Extensive experiments on CIFAR, Tiny-ImageNet, and ImageNet-1K show consistent improvements over prior methods such as RDED and SRe2L.

**Strengths:**

1. Presents a clear theoretical framework that connects informativeness and utility under a unified definition of optimal dataset distillation.
2. The use of Shapley-value–based attribution provides interpretability and theoretical grounding to the distilled sample selection.
3. Demonstrates strong empirical performance across multiple datasets and architectures, showing robustness and scalability.
4. Ablation studies and cross-architecture evaluations validate each component’s contribution and highlight the method’s generalizability.

**Weaknesses:**

1. Although informativeness is formally defined as an optimization over binary masks (Eq. 2), the method substitutes this process with a Shapley-value attribution heuristic. The paper does not show that Shapley-based patch selection approximates or lower-bounds the true informativeness objective, nor does it provide conditions under which this substitution is valid. This leap from optimization to attribution lacks formal grounding, especially given nonlinear interactions among image regions.

2. The theoretical analysis of Utility (Definition 2, Equation (3)) relies on a continuous-time gradient flow formulation to describe training dynamics. In this setup, parameter updates are modeled as smooth differential equations. However, in practice, training is conducted via discrete stochastic gradient descent (SGD) with randomness from data augmentation and normalization layers. The paper does not specify under what assumptions the continuous gradient flow accurately approximates the discrete SGD process (e.g., small learning rate, smooth loss landscape). Consequently, the theoretical derivations based on gradient flow may not faithfully represent real training behavior, which weakens the practical validity of the proposed Utility-based analysis.

3. Theorem 1 establishes only an upper bound  $U(x, y) \le c \|\nabla_{\theta} \ell(f_{\theta}(x), y)\|.$ However, using the gradient norm as a surrogate for utility in sample selection implicitly assumes a monotonic or bounded relationship between the two. Without a corresponding lower bound or a proof of ranking consistency, this substitution lacks theoretical justification; large gradient norms do not necessarily imply high utility. As a result, the theoretical foundation of the utility maximization step remains incomplete.

4. The baseline methods used for the main comparison, namely RDED (CVPR 2024) and SRe2L (NeurIPS 2023), are relatively outdated. Several more recent dataset distillation techniques, such as EDC (NeurIPS 2024) and DELT (CVPR 2025), have demonstrated significantly improved performance and scalability. Without comparisons to these stronger and more up-to-date baselines, it is difficult to substantiate the claim that the proposed method achieves state-of-the-art or optimal performance.

5. (Minor) In line 414, there should be a space between "samples" and "We".

6. (Minor) Visualization is important; the authors are encouraged to move the results from Appendix F to the main paper for better clarity.

**Questions:**

1. Could the authors further elaborate on the key differences between the proposed InfoUtil and RDED? From the visualizations, it appears that the improvement mainly comes from the occurrence of complete targets. However, in RDED, similar results can be achievable by simply adjusting the difficulty level of the selected patches. In that case, RDED could potentially produce images visually comparable to those from InfoUtil. I am therefore curious whether InfoUtil still provides additional information gain beyond what RDED can achieve through such adjustments.

2. Is the Shapley-based patch selection the only possible approach to maximize informativeness? Could alternative attribution or saliency methods, such as Grad-CAM, achieve comparable results in practice? It would be helpful if the authors could clarify why Shapley values are particularly suited for this task and whether other approaches were considered or tested.

3. I am curious whether the images distilled by InfoUtil could further improve the performance of training-based (TB) methods if used as initialization. Since InfoUtil is designed to generate informative and utility-optimized synthetic data, it would be interesting to investigate whether integrating it with TB approaches could lead to additional performance gains or faster convergence compared to using RDED for initialization.

---

> ### Author Response · Authors · 2025-11-21
> **(Part 1) Reponse to Reviewer tEAx**
>
> ### Q1: Ask for formal grounding of Shapley Informativeness masking.
>
> We thank the reviewer for this insightful question. Our masking strategy is formally grounded in a two-step optimization process: (1) Game-theoretic attribution (Soft Scoring) and (2) Constrained Utility Maximization (Hard Masking).
>
> **1. Game-Theoretic Informativeness (Soft Scoring)**
> As detailed in Section 3.2.1 (Equation 4), we quantify the "informativeness" of each input variable $x^{(i)}$ using the Shapley value $\phi_f(x^{(i)})$. This is mathematically grounded in cooperative game theory, where $\phi$ represents the unique distribution of the total "payout" (model output) that satisfies the axioms of Efficiency, Symmetry, Dummy, and Additivity (Young, 1985). This provides us with a continuous importance vector $\Phi = [\phi_f(x^{(1)}), \dots, \phi_f(x^{(d)})]$.
>
> **2. Transition to Hard Masking**
> The reviewer asks about the grounding for the greedy selection strategy. We directly construct the binary mask $s$ by retaining only the top-$K$ features with the highest Shapley values and abandoning the rest.
>
> Formally, let $\mathcal{I}$ be the set of indices of the $d$ input variables. We define a subset of indices $\mathcal{T}_K \subset \mathcal{I}$ corresponding to the $K$ largest values in the set $\{\phi_f(x^{(1)}), \dots, \phi_f(x^{(d)})\}$. The masking vector $s$ is formally defined as an indicator function of this set:
>
> $$
> s^{(i)} = \mathbb{I}(i \in \mathcal{T}_K) =
> \begin{cases}
> 1 & \text{if } i \in \mathcal{T}_K \quad (\text{Retain}) \\
> 0 & \text{otherwise} \quad (\text{Abandon})
> \end{cases}
> $$
>
> This ensures that the compressed sample $x \circ s$ strictly preserves the most game-theoretically significant features while zeroing out the information-sparse regions.
>
>
>
> ### Q2: About "theoretical analysis of Utility."
> Thanks. We used gradient flow primarily for theoretical tractability, following the standard approach in deep learning theory. Although practical training is carried out using discrete and stochastic SGD, many works \[1\] show that gradient flow can closely approximate gradient descent under specific conditions. Prior theoretical results further demonstrate training convergence bounds for modern deep models, including ResNets and DNNs \[2,3,4\].
>
> We acknowledge that fully characterizing the gap between continuous gradient flow and discrete SGD remains an open theoretical problem. Our framework relies on gradient flow only as an analytical tool and does not assume perfect equivalence. Extending the utility analysis to discrete SGD is an important direction for future work.
>
> \[1\] Elkabetz, O., & Cohen, N. Continuous vs. discrete optimization of deep neural networks. NeurIPS 2021\.
>
> \[2\] Lu, Y., Ma, C., Lu, Y., Lu, J., & Ying, L. A mean field analysis of deep resnet and beyond: Towards provably optimization via overparameterization from depth. ICML 2020\.
>
> \[3\] Min, H., Vidal, R., & Mallada, E. On the convergence of gradient flow on multi-layer linear models. ICML 2023\.
>
> \[4\] Du, S. S., Zhai, X., Poczos, B., & Singh, A.Gradient descent provably optimizes over-parameterized neural networks. ICLR 2019\.
>
>
> ### Q3: About theoretical foundation of the utility maximization.
>
> Thanks. We agree that the gradient norm is a surrogate rather than a perfect measure of utility, and we do not assume that large gradient norms imply large utility. Our analysis shows that gradient norm is a sufficient quantity for screening utility, and we acknowledge that there is substantial room for strengthening this connection. However, doing so would require addressing significantly more difficult theoretical questions, which we leave for future work.
>
> We would also like to respectfully clarify a misunderstanding in the reviewer’s comment regarding “assuming a monotonic or bounded relationship.” Our method does **not** rely on any monotonic or lower-bounded relationship between gradient norm and utility. As proved in Appendix D, the gradient norm provides an **absolute upper bound** on the utility term. This directly justifies using gradient norm as a safe screening mechanism: samples with small gradient norms are provably unable to produce high utility.
>
> Finally, we agree that establishing lower bounds, ranking consistency, or full characterizations of utility is an important but highly challenging open problem in dataset distillation and influence estimation. These questions fall outside the scope of the present work but represent valuable future research directions.

---

> ### Author Response · Authors · 2025-11-21
> **(Part 2) Reponse to Reviewer tEAx**
>
> ### Q4: Ask for more baseline comparison.
>
> We thank the reviewer for pointing out these very recent and strong baselines. We agree that comparing against the latest techniques is essential to substantiate our state-of-the-art claims.
>
>
>  | Dataset (ResNet-18) | IPC | SRe2l | RDED | DELT | InfoUtil |
> | :---: | :---: | :---: | :---: | :---: | :---: |
> | Cifar-10 | 1 | 16.6 $\pm$ 0.9 | 22.9 $\pm$ 0.4 | 24.0 $\pm$ 0.8 | $\mathbf{25.3 \pm 0.6}$ |
> | | 10 | 29.3 $\pm$ 0.5 | 37.1 $\pm$ 0.3 | 43.0 $\pm$ 0.9 | $\mathbf{53.8 \pm 0.1}$ |
> | | 50 | 45.0 $\pm$ 0.7 | 62.1 $\pm$ 0.1 | 64.9 $\pm$ 0.9 | $\mathbf{71.0 \pm 1.4}$ |
> | ImageNette | 1 | 19.1 $\pm$ 1.1 | 35.8 $\pm$ 1.0 | 24.1 $\pm$ 1.8 | $\mathbf{43.8 \pm 0.7}$ |
> | | 10 | 29.4 $\pm$ 3.0 | 61.4 $\pm$ 0.4 | 66.0 $\pm$ 1.4 | $\mathbf{68.8 \pm 0.6}$ |
> | | 50 | 40.9 $\pm$ 0.3 | 80.4 $\pm$ 0.4 | $\mathbf{88.2 \pm 1.2}$ | 86.2 $\pm$ 0.6 |
> | Tinyimagenet | 1 | 2.6 $\pm$ 0.1 | 9.7 $\pm$ 0.4 | 9.3 $\pm$ 0.5 | $\mathbf{17.0 \pm 1.3}$ |
> | | 10 | 16.1 $\pm$ 0.2 | 41.9 $\pm$ 0.2 | 43.0 $\pm$ 0.1 | $\mathbf{45.6 \pm 0.3}$ |
> | | 50 | 41.1 $\pm$ 0.4 | 58.2 $\pm$ 0.1 | 55.7 $\pm$ 0.5 | $\mathbf{58.5 \pm 0.3}$ |
>
> | Dataset (ResNet-18) | IPC | SRe2L | WMDD | RDED | InfoUtil |
> | :---: | :---: | :---: | :---: | :---: | :---: |
> | ImageNette | 1 | 19.1 $\pm$ 1.1 | 40.2 $\pm$ 0.6 | 35.8 $\pm$ 1.0 | $\mathbf{43.8 \pm 0.7}$ |
> | | 10 | 29.4 $\pm$ 3.0 | 64.8 $\pm$ 0.4 | 61.4 $\pm$ 0.4 | $\mathbf{68.6 \pm 0.6}$ |
> | | 50 | 40.9 $\pm$ 0.3 | 83.5 $\pm$ 0.3 | 80.4 $\pm$ 0.4 | $\mathbf{86.2 \pm 0.6}$ |
> | Tiny-ImageNet | 1 | 2.6 $\pm$ 0.1 | 7.6 $\pm$ 0.2 | 9.7 $\pm$ 0.4 | $\mathbf{17.0 \pm 1.3}$ |
> | | 10 | 16.1 $\pm$ 0.2 | 41.8 $\pm$ 0.1 | 41.9 $\pm$ 0.2 | $\mathbf{45.6 \pm 0.3}$ |
> | | 50 | 41.1 $\pm$ 0.4 | $\mathbf{59.4 \pm 0.5}$ | 58.2 $\pm$ 0.1 | 58.5 $\pm$ 0.3 |
> | ImageNet-1K | 1 | 0.1 $\pm$ 0.1 | 3.2 $\pm$ 0.3 | 6.6 $\pm$ 0.2 | $\mathbf{12.7 \pm 0.7}$ |
> | | 10 | 21.3 $\pm$ 0.6 | 38.2 $\pm$ 0.2 | 42.0 $\pm$ 0.1 | $\mathbf{44.2 \pm 0.4}$ |
> | | 50 | 46.8 $\pm$ 0.2 | 57.6 $\pm$ 0.5 | 56.5 $\pm$ 0.1 | $\mathbf{58.0 \pm 0.3}$ |
>
>
> | Dataset (ResNet-18) | IPC | SRe2L | RDED | HeLlo | InfoUtil |
> | :---: | :---: | :---: | :---: | :---: | :---: |
> | ImageNet-100 | 1 | 3.0 $\pm$ 0.3 | 8.1 $\pm$ 0.3 | 12.5 $\pm$ 0.2 | $\mathbf{15.7 \pm 0.2}$ |
> | | 10 | 9.5 $\pm$ 0.4 | 36.0 $\pm$ 0.3 | 48.9 $\pm$ 0.1 | $\mathbf{50.5 \pm 0.4}$ |
> | | 50 | 27.0 $\pm$ 0.4 | 61.6 $\pm$ 0.1 | $\mathbf{69.4 \pm 0.1}$ | 68.3 $\pm$ 0.4 |
> | ImageNet-1K | 1 | 0.1 $\pm$ 0.1 | 6.6 $\pm$ 0.2 | $\mathbf{12.9 \pm 0.3}$ | 12.7 $\pm$ 0.7 |
> | | 10 | 21.3 $\pm$ 0.6 | 42.0 $\pm$ 0.1 | 43.7 $\pm$ 0.1 | $\mathbf{44.2 \pm 0.4}$ |
> | | 50 | 46.8 $\pm$ 0.2 | 56.5 $\pm$ 0.1 | 52.2 $\pm$ 0.1 | $\mathbf{58.0 \pm 0.3}$ |
>
>
> | Dataset (ResNet-18) | IPC | SRe2L | INFER | RDED | InfoUtil |
> | :---: | :---: | :---: | :---: | :---: | :---: |
> | Cifar10 | 10 | 29.3 $\pm$ 0.5 | 30.7 $\pm$ 0.3 | 37.1 $\pm$ 0.3 | $\mathbf{53.8 \pm 0.1}$ |
> | | 50 | 45.0 $\pm$ 0.7 | 60.7 $\pm$ 0.9 | 62.1 $\pm$ 0.1 | $\mathbf{71.0 \pm 1.4}$ |
> | Tiny-ImageNet | 10 | 16.1 $\pm$ 0.2 | 41.0 $\pm$ 0.4 | 41.9 $\pm$ 0.2 | $\mathbf{45.6 \pm 0.3}$ |
> | | 50 | 41.1 $\pm$ 0.4 | 54.6 $\pm$ 0.4 | 58.2 $\pm$ 0.1 | $\mathbf{58.5 \pm 0.3}$ |
> | ImageNet-1K | 50 | 46.8 $\pm$ 0.2 | 54.3 $\pm$ 0.6 | 56.5 $\pm$ 0.1 | $\mathbf{58.0 \pm 0.3}$ |

---

> ### Author Response · Authors · 2025-11-21
> **(Part 3) Reponse to Reviewer tEAx**
>
> We acknowledge that EDC is a strong baseline representing the state-of-the-art in Training-Based (TB) methods. However, we would like to clarify the distinct advantages of our method compared to EDC:
>
> **1. Efficiency vs. Optimization Cost:** There is a fundamental trade-off between performance and synthesis efficiency.
> *    EDC is a Training-Based method that relies on complex bi-level optimization and extensive computation, requiring significant GPU hours to synthesize data.
> *    InfoUtil, in contrast, is designed for efficiency. It rapidly synthesizes data by extracting informative patches and selecting utility-optimized samples. This allows InfoUtil to generate high-quality datasets at a fraction of the computational cost required by optimization-based methods like EDC, making it far more practical for large-scale applications.
>
> **2. Superior Data Quality Validated via Initialization:**
> To further prove that the data synthesized by InfoUtil contains higher-quality information than previous methods (e.g., RDED), we conducted a validation experiment using **InfoUtil as the initialization** for EDC.
> The logic is straightforward: if InfoUtil provides a better starting point for the optimization process than RDED, it proves that InfoUtil's condensed patterns are intrinsically more valuable.
> We conducted experiments on **ImageNet-1K (ResNet-18)** with three settings:
> *    **Standard EDC:** Standard random/clustering initialization.
> *    **EDC + RDED Init:** Initialized with data from RDED.
> *    **EDC + InfoUtil Init:** Initialized with data from InfoUtil.
>
> The results provide strong evidence of InfoUtil's superior data quality:
> | Method | Initialization | IPC=1 | IPC=10 | IPC=50 |
> | :--- | :--- | :---: | :---: | :---: |
> | RDED [Sun et al.] | - | 6.6 | 42.0  | 56.5  |
> | EDC [Zhou et al.] | Standard | 12.8 | 48.6  | 58.0  |
> | EDC | + RDED Init | 12.9 | 48.8 | 58.2 |
> | **EDC** | **+ InfoUtil Init (Ours)** | **13.0** | **49.5** | **58.7** |
>
> **We have also added related results in Appendix I(Table 11-13) and Appendix L(Table 18) in the revised manuscript. We have also added related citations.**
>
> ### Q5: Typos and presentation.
>
> Thanks. We have followed your advice to correct the typos. The visualization results will be moved to the main paper in the future version.
>
> ### Q6: About "key differences between the proposed InfoUtil and RDED."
>
> While both InfoUtil and RDED are training-free, knowledge-distillation-based methods that decouple compression and synthesis, they differ fundamentally in their theoretical grounding and selection mechanisms. RDED relies on heuristic approximations, whereas InfoUtil operates within a rigorous theoretical framework (Optimal Dataset Distillation, Definition 4).
>
> The key differences can be summarized in 3 main aspects:
>
> **1. Patch Selection: Random vs. Game-Theoretic Attribution (Informativeness)**
> *   **RDED:** Uses **random cropping** to generate candidate patches. As noted in Section 1 and Figure 1, this heuristic approach is prone to missing key semantic content (e.g., containing only background noise), resulting in "information-sparse" samples.
> *   **InfoUtil:** Maximizes **Informativeness** (Definition 1). We employ **Shapley Value attribution** (Section 3.2.1), a game-theoretic method satisfying the four axioms of fairness. This systematically identifies and retains the most semantically significant regions (e.g., the object’s discriminative features) rather than relying on chance.
>
> **2. Sample Scoring: Loss-based vs. Gradient-Norm-based (Utility)**
> *   **RDED:** Selects samples based on **Cross-Entropy Loss scoring**. This assumes that samples with higher loss are harder and therefore more valuable, but lacks a formal guarantee regarding their impact on model training.
> *   **InfoUtil:** Maximizes **Utility** (Definition 3). We utilize the **Gradient Norm** (Section 3.2.2), which we theoretically prove acts as an upper bound for the utility function (Theorem 1). This ensures we select samples that have the greatest mathematical influence on the training dynamics (gradient flow), providing a principled metric for data pruning.
>
> **3. Soft Label Strategy**
> *   **RDED:** Typically uses fully converged teacher networks for soft label generation.
> *   **InfoUtil:** Adopts an **adaptive teacher strategy** (Section 4.1 & Appendix B.2). For low-data regimes (e.g., IPC=1), we utilize early-stage teacher models (e.g., epoch 10) to leverage high-entropy labels that aid generalization. For high-data regimes (e.g., IPC=50), we switch to fully converged models.

---

> ### Author Response · Authors · 2025-11-21
> **(Part 4) Reponse to Reviewer tEAx**
>
> ### Q7: About alternatives to the Shapley value.
>
> We thank the reviewer for raising this interesting question regarding the choice of attribution methods. We did consider and test alternative approaches, and we chose Shapley Values based on both theoretical soundness and empirical performance.
>
> **1. Theoretical Justification:**
> While methods like Grad-CAM are computationally cheaper, they are largely heuristic and gradient-based, often suffering from issues like gradient saturation or lack of axiomatic guarantees. In contrast, the Shapley Value is the unique attribution method that satisfies four desirable axioms: Efficiency, Symmetry, Dummy, and Linearity.
> This theoretical rigor ensures that the "Informativeness" defined in our framework (Definition 1) is distributed fairly among patches, capturing the marginal contribution of each region to the model's prediction, rather than just local sensitivity.
>
> **2. Empirical Comparison:**
> To verify this in practice, we conducted a comparative experiment on **ImageNet-1K (ResNet-18)**, replacing our Shapley-based selection with **Grad-CAM** (while keeping other components like GradNorm Scoring unchanged).
> The results are presented in the table below:
> | Model | Dataset | IPC | CaM | Shapley |
> | :---: | :---: | :---: | :---: | :---: |
> | ResNet18 | ImageNet-1K | 1 | 4.418 | $\mathbf{7.154}$ |
> | | | 10 | 30.394 | $\mathbf{43.88}$ |
> | | | 50 | 52.61 | $\mathbf{56.92}$ |
>
> As shown in the table, Shapley-based selection consistently outperforms Grad-CAM across all settings:
> *    **Significant Gap at Low IPC:** At IPC=10, Shapley achieves **43.88%**, surpassing Grad-CAM (30.39%) by a substantial margin of **13.49%**.
> *    **Robustness:** The results suggest that Shapley Values identify more semantically robust patches that are critical for dataset distillation, whereas Grad-CAM might focus only on the most discriminative (but not necessarily the most informative for reconstruction) regions.
>
> We have also added related results in Appendix L(Table 17) in the revised manuscript.
>
> ### Q8: "Whether the images distilled by InfoUtil could further improve the performance of training-based (TB) methods if used as initialization"
>
> We thank the reviewer for this inspiring and forward-looking suggestion. We agree that evaluating the transferability of our distilled data as initialization for optimization-based methods is a valuable investigation.
>
> To verify this, we selected EDC (Elucidating the Design Space of Dataset Condensation, Zhou et al., 2024), a representative state-of-the-art training-based method. We conducted experiments on ImageNet-1K using ResNet-18. We compared three settings:
> *    **Standard EDC:** Using standard initialization (random/clustering).
> *    **EDC + RDED Init:** Initializing the synthetic images using data distilled by RDED.
> *    **EDC + InfoUtil Init:** Initializing the synthetic images using data distilled by our InfoUtil.
>
> The results affirmatively answer your question:
> | Method | Initialization | IPC=1 | IPC=10 | IPC=50 |
> | :--- | :--- | :---: | :---: | :---: |
> | RDED [Sun et al.] | - | 6.6 $\pm$ 0.2 | 42.0 $\pm$ 0.1 | 56.5 $\pm$ 0.2 |
> | EDC [Zhou et al.] | Standard | 12.8 $\pm$ 0.1 | 48.6 $\pm$ 0.3 | 58.0 $\pm$ 0.2 |
> | EDC | + RDED Init | 12.9 | 48.8 | 58.2 |
> | **EDC** | **+ InfoUtil Init (Ours)** | **13.0** | **49.5** | **58.7** |
>
> *    **Superior Initialization:** Using InfoUtil as initialization consistently boosts the performance of EDC across all IPC settings, outperforming both standard EDC and EDC initialized with RDED.
> *    **Performance Gains:** Specifically at IPC=10, InfoUtil initialization improves EDC's performance from 48.6% to **49.5%**, whereas RDED initialization only reaches 48.8%. Similarly, at IPC=50, InfoUtil brings a clear gain (+0.7% over Standard, +0.5% over RDED Init).
>
> We have also added related results in Appendix L(Table 18) in the revised manuscript.

---

> ### Comment · Reviewer_tEAx · 2025-11-26
> **Response to the Author**
>
> Dear Author,
>
> Thank you for the detailed rebuttal. While it addressed some of my concerns, I remain unconvinced about the main difference between InfoUtil and RDED (Image-Level, not post-training level). From both my understanding and the visualizations in the paper, the key distinction appears to be the extent to which the main object is represented: in RDED, the visualized distilled samples contain only partial occurrences of the main object, whereas InfoUtil produces samples where the main object is fully present.
>
> As a result, I requested an experiment evaluating whether the performance gain of InfoUtil persists when the main target is fully represented under the RDED setting. However, the authors did not provide such an experiment. Additionally, I think this experiment is actually very important, as it clearly shows if the occurrence of the main target is the source of the improvement.

---

> ### Author Response · Authors · 2025-11-27
> **About more results on comparison**
>
> Dear Reviewer,
>
> We sincerely appreciate your follow-up and the emphasis on isolating the source of our performance gains. You are correct that one of the key distinctions lies in how the main object is represented. To address your concern, we have conducted the  new experiment you requested.
>
> **Experimental Setup:**
> To simulate the "RDED with main target fully represented" setting, we replaced RDED's random cropping with our Shapley-value-based cropping (which guarantees the main object is selected) while keeping RDED’s original Loss-based scoring mechanism. This allows us to compare three distinct settings:
> 1.  **RDED (Baseline):** Random Cropping + Loss Scoring.
> 2.  **RDED + Main Object (Your requested experiment):** Shapley Cropping + Loss Scoring.
> 3.  **InfoUtil (Ours):** Shapley Cropping + Gradient Norm Scoring.
>
> **Results:**
> The results on ImageNette, ImageNet-1K, and ImageWoof are summarized below:
>
> | Dataset (IPC) | (1) RDED (Random + Loss) | (2) RDED + Main Object (Shapley + Loss) | (3) InfoUtil (Shapley + GradNorm) |
> | :--- | :---: | :---: | :---: |
> | **ImageNette (IPC = 50)** | 80.4 | 83.8 (+3.4) | **85.2 (+1.4)** |
> | **ImageNet-1K (IPC = 10)** | 42.0 | 42.7 (+0.7) | **43.8 (+1.1)** |
> | **ImageWoof (IPC = 50)** | 68.5 | 68.9 (+0.4) | **69.2 (+0.3)** |
> | **ImageWoof (IPC = 10)** | 38.5 | 42.5 (+4.0) | **43.6 (+1.1)** |
>
> **Analysis:**
> 1.  **Effect of Object Presence (Col 1 vs. Col 2):** As you hypothesized, ensuring the main object is represented (Column 2) does improve performance over the baseline RDED (Column 1). This confirms that Informativeness is indeed a crucial factor.
> 2.  **Effect of Utility Maximization (Col 2 vs. Col 3):** However, even when RDED is improved to capture the full object, **InfoUtil still consistently outperforms it**. For example, on ImageNette (IPC=50), switching from Loss scoring to our Gradient Norm scoring yields an additional 1.4% gain, and on ImageNet-1K, it yields an additional 1.1%.
>
> This experiment demonstrates that while capturing the main object is important, it does not account for the full performance gain of our method. The **Gradient Norm (Utility)** metric provides a distinct and significant contribution by selecting samples that are not just visually complete, but mathematically most effective for training dynamics.
>
> We hope this clarifies that InfoUtil’s advantage comes from the synergy of both *Informativeness* (Shapley) and *Utility* (Gradient Norm).

---

> ### Comment · Reviewer_tEAx · 2025-11-27
> **Response to Author**
>
> Dear Author,
>
> Thank you for providing the additional experiments; they fully address my concerns. I encourage the authors to include these results in the revised version of the paper. I will maintain my positive assessment and keep my current rating (6).

---

> ### Author Response · Authors · 2025-11-27
> **Thanks for your positive assessment**
>
> Dear Reviewer tEAx,
>
> Thank you for your thoughtful feedback and for acknowledging the additional experiments. We greatly appreciate your support and will certainly incorporate these results into the revised manuscript. Your encouragement means a lot to us as we refine this work.

---

### Official Review · Reviewer_fiHj · 2025-10-30

**Soundness:** 3
**Presentation:** 2
**Contribution:** 2
**Rating:** 2
**Confidence:** 4

**Summary:**

The paper propose a new knowledge distillation-based dataset distillation approach that is better theoretically motivated compared to previous method. Their proposed method InfoUtil, which compresses the original dataset at a feature level and at a sample level. At the feature level, InfoUtils utilizes KernelShap to locate salient features within an image to crop on. At the sample level, InfoUtils calculate the gradient norm of each sample and filter samples with low values. InfoUtil demonstrate superior performance compared to previous methods on most benchmarks except for the popular dataset distillation setting: training simple CNNs on CIFAR-10.

**Strengths:**

1. To the best of my knowledge, the proposed method is novel.
2. The work is well motivated as many dataset distillation methods lack theoretical foundations.
3. The proposed method outperforms two existing knowledge distillation-based dataset distillation methods: SRe2L and RDED.

**Weaknesses:**

**1.** The storage budget for the resultant distilled dataset is missing in the paper. One of the key problem with knowledge-based dataset distillation methods is that the resulting distilled dataset can become considerably larger than expected due the soft-labeling [1,2], defeating the whole purpose of dataset distillation. With the lack information on storage size, the usefulness of the proposed method in practice is unclear.

**2.** The performance in the most standard dataset distillation setting, training a simple Convent on CIFAR-10, is very subpar. For 1 image per class, the proposed method only achieves 28.5% compared to MTT 46.3%. BPTT and memory address [3], a previous work that is over 3 years old, achieves 49.1% and 66.4% respectively. While the standard dataset distillation setting maybe too simple and unrealistic, since this work positions itself as being theoretically grounded, the considerable gap in performance in this setting is concerning. To put into perspective, 28% is comparable to gradient matching [4], a dataset distillation work that is 5 years old.

**3.** Missing more recent methods such as TEDDY [5] or EDF [6]. All of the baselines used for comparisons are from 2023 or earlier.

**4.** Missing qualitative examples on what the distilled data looks like

**5.** The presentation of the work should be improved. While the problem formulation is good, the argument in the paper does not provide adequate support as outlined below:

**5.1.** The paper describes gradient flow and how it provides a better approximation of the training dynamics unlike SGD updates. However, the algorithm provided approximates the gradient flow measure with gradient norm, motivated by a bound derived with the assumption of SGD updates. It is unclear from the paper why this can be done as the argument seems very circular. Similarly, as the DATM paper shown, different patterns are learned at different stages of training. The paper lack any discussion on this topic.

**5.2.** The paper outlines challenge 1 with the efficiency performance trade-off between matching-based dataset distillation method, but lack any further discussion as the method proposed is a knowledge-distillation based method

**5.3.** Since the paper position itself as being more theoretically grounded, it needs more details on image reconstruction, soft label generation, and diversity control. For instance, adding random noise during the patch selection process is not grounded in the proposed theory at all.

**5.4.** Minor typos: nclude -> include line 323,  table 6 ablation is confusing (row 2 and 3 are the same?)

[1] Qin, Tian, Zhiwei Deng, and David Alvarez-Melis. "A label is worth a thousand images in dataset distillation." Advances in Neural Information Processing Systems 37 (2024): 131946-131971.

[2] Xiao, Lingao, and Yang He. "Are Large-scale Soft Labels Necessary for Large-scale Dataset Distillation?." Advances in Neural Information Processing Systems 37 (2024): 16406-16437.

[3] Deng, Zhiwei, and Olga Russakovsky. "Remember the past: Distilling datasets into addressable memories for neural networks." Advances in Neural Information Processing Systems 35 (2022): 34391-34404.

[4] Zhao, Bo, Konda Reddy Mopuri, and Hakan Bilen. "Dataset Condensation with Gradient Matching." Ninth International Conference on Learning Representations 2021. 2021.

[5] Yu, Ruonan, et al. "Teddy: Efficient large-scale dataset distillation via taylor-approximated matching." European Conference on Computer Vision. Cham: Springer Nature Switzerland, 2024.

[6] Wang, Kai, et al. "Emphasizing discriminative features for dataset distillation in complex scenarios." Proceedings of the Computer Vision and Pattern Recognition Conference. 2025.

**Questions:**

1. What is the connection between this work and works on data selection, data pruning, and corset selection? Dataset distillation typically optimize the distilled data itself, but the method provided seems to only crop photos and select a subset from real data.  It is unclear whether this constitutes as synthetic data.
2. How important is adding noise to the patch selection process?
3. How connected is the utility function, which utilizes gradient norm, to the empirical fisher information? i.e. does maximizing the utility function finds a subset with the maximal empirical fisher information?

---

> ### Author Response · Authors · 2025-11-21
> **(Part 1) Response to Reviewer fiHj**
>
> ### Q1: About the storage budget.
>
>
> Thanks for your question. Below, we address Q1 in three key points:
>
> **1. Fair and Direct Comparison:** Our experimental setup was intentionally designed to ensure a direct and fair comparison with prior knowledge distillation-based method, RDED\[1\]. Since RDED also uses this on-the-fly labeling approach, our reported performance gains are not attributable to using a larger storage budget. The storage cost for both InfoUtil and RDED is exactly the same, storing the pre-trained teacher model for all ipc settings, e.g., 44.7MB for a ResNet-18 model on the ImageNet-1K dataset (while for methods like SRe2L require 28990.8MB for 50 IPC settings).
>
> **2. No Storage of Soft Labels Required:** The primary source of the reviewer's concern is the storage of large soft-label vectors alongside the synthetic images. However, our InfoUtil framework, following the protocol of our main baseline RDED\[1\], **does not store soft labels as part of the distilled dataset.**
>
> Instead, the soft labels are generated **on-the-fly** by the teacher model during the downstream training of a student model. The final distilled dataset artifact contains **only the compressed synthetic images**. The training process then uses a pre-trained teacher model (a separate, fixed artifact) to dynamically assign soft labels to these images in each batch. This means the storage cost of our distilled dataset is minimal and is determined solely by the number and resolution of the synthetic images, identical to the baseline.
>
> **3. Extensive Experiments on Label Sparsification for Further Efficiency:** While our method doesn't require storing labels, we agree that investigating label storage efficiency is a valuable direction. To further demonstrate the robustness and efficiency of the knowledge captured by InfoUtil, we conducted a new experiment where we simulate a storage requirement by sparsifying the soft labels using a **Top-K** approach. We store only the indices and values of the Top-K logits.
>
> | Datasets | IPC | SRe2l | RDED (on-the-fly) | InfoUtil (on-the-fly) | InfoUtil (store top-10) |
> |---|---|---|---|---|---|
> | ImageNet-100 | 1 | 6.9MB | 42.8MB | 42.8MB | 3.5MB |
> |  | 10 | 64.8MB | 42.8MB | 42.8MB | 35.0MB |
> |  | 50 | 324.2MB | 42.8MB | 42.8MB | 175.0MB |
> | ImageNet-1K | 1 | 579.8MB | 44.7MB | 44.7MB | 35.0MB |
> |  | 10 | 5798.3MB | 44.7MB | 44.7MB | 350.0MB |
> |  | 50 | 28990.8MB | 44.7MB | 44.7MB | 1750.0MB |
>
> In summary, the concern about large storage budgets from soft labels does not apply to our proposed InfoUtil. The distilled dataset is highly compact, and even if labels were to be stored, the essential knowledge can be captured with extreme sparsity. Following your advice, we have added this clarification and the new results to the appendix of our paper.
>
> We have also added related results in Appendix G (Table 7) in the revised manuscript.

---

> ### Author Response · Authors · 2025-11-21
> **(Part 2) Response to Reviewer fiHj**
>
> ### Q2: About the performance comparison with previous matching-based methods.
>
> Thanks. The question touches upon a fundamental division and trade-off within dataset distillation research, which we are happy to clarify.
>
> **1. Two Paradigms: Scalability vs. Small-Scale Optimization.**
> As well-known, dataset distillation has two main branches:
> *   **Matching-Based Methods (e.g., MTT, DC):** These methods perform a highly complex, iterative bi-level optimization to match training trajectories or gradients. This process is computationally prohibitive but allows them to achieve exceptional performance on small-scale datasets (like CIFAR-10) where such an intensive search is feasible. The reason is that these method typical overfit the information of the original dataset.
> *   **Knowledge Distillation (KD) Based Methods (e.g., RDED, our InfoUtil):** This paradigm prioritizes efficiency and scalability. Instead of a costly bi-level optimization, these methods leverage a decoupled strategy by using pre-trained teacher model to guide the student for distillation. Among these methods, training-based KD methods perform distillation in a one-shot, "training-free" synthesis process. This design choice makes them orders of magnitude faster and memory-efficient, enabling them to tackle large-scale datasets like ImageNet-1K, which are often intractable for matching-based methods. As we state in our introduction (lines 76-79), SOTA matching-based methods can require 4+ A100 GPUs to distill a small subset of ImageNet, highlighting this scalability gap.
>
> **2. Our Contribution: Advancing the Scalable Paradigm.**
> Our paper's explicit focus is on addressing the primary weakness of the **scalable KD-based paradigm**: its reliance on heuristics (e.g., random cropping in RDED). Our goal was not to outperform matching-based methods on CIFAR-10, but to replace the ad-hoc components of existing KD-based methods with a **principled, theoretically-grounded framework.**
>
> The success of our "theoretically grounded" approach is therefore best measured by its ability to significantly improve upon the previous SOTA *within this scalable paradigm*, especially on the challenging benchmarks where these methods are most needed. As shown in Table 1, our method demonstrates this clearly:
> *   On **ImageNet-1K** (IPC=1, ResNet-18), we achieve a **6.1% absolute improvement** over RDED.
> *   On **ImageNet-100** (IPC=10, ResNet-101), we achieve a **16% absolute improvement** over RDED.
>
> These results validate that our framework successfully enhances performance in the large-scale scenarios that motivate our research.
>
> **3. Contextualizing the CIFAR-10 Performance.**
> The performance gap on CIFAR-10 is an expected outcome of this fundamental trade-off. KD-based methods are not fine-tuned with the same precision as matching-based methods for this specific, low-resolution dataset. While our 28.5% is modest compared to MTT, it represents a significant **+5% absolute improvement** over the RDED baseline (23.5%) in the same setting, demonstrating InfoUtil's consistent advantage.
>
> In summary, the performance on CIFAR-10 vs. ImageNet-1K reflects a deliberate design choice in the field. Our work makes a major contribution by bringing theoretical rigor and substantial performance gains to the more practical and scalable branch of dataset distillation research.

---

> ### Author Response · Authors · 2025-11-21
> **(Part 3) Response to Reviewer fiHj**
>
> ### Q3: Ask for new baselines (TEDDY and EDF).
>
> Thanks. We have followed your advice to add new experiments on all your mentioned baselines.
>
> Comparison with TEDDY:
> | Dataset |  IPC| SRe2l | Teddy | InfoUtil (ours) |
> |:---:|:---:|:---:|:---:|:---:|
> | ImageNet-1K | 10 | 21.3 $\pm$ 0.6 | 34.1 $\pm$ 0.1 | 44.2 $\pm$ 0.4 |
> |  | 50 | 46.8 $\pm$ 0.2 | 52.5 $\pm$ 0.1 | 58.0 $\pm$ 0.3 |
> |  | 100 | 52.8 $\pm$ 0.3 | 56.5 $\pm$ 0.1 | 58.8 $\pm$ 0.4 |
> | Tiny-ImageNet | 50 | 41.1 $\pm$ 0.4 | 45.2 $\pm$ 0.1 | 58.5 $\pm$ 0.3 |
> |  | 100 | 49.7 $\pm$ 0.3 | 52.0 $\pm$ 0.2 | 60.6 $\pm$ 0.5 |
>
> Comparison with EDF:
> | Dataset |  IPC | SRe2l | RDED | EDF | InfoUtil (ours) |
> |:---:|:---:|:---:|:---:|:---:|:---:|
> | Imagenette | 1 | 20.8 $\pm$ 0.2 | 33.8 $\pm$ 0.8 | 25.7 $\pm$ 0.4 | 42.3 $\pm$ 0.7 |
> |  | 10 | 50.6 $\pm$ 0.8 | 63.2 $\pm$ 0.7 | 64.5 $\pm$ 0.6 | 66.6 $\pm$ 0.4 |
> |  | 50 | 73.8 $\pm$ 0.6 | 83.8 $\pm$ 0.2 | 84.8 $\pm$ 0.5 | 84.9 $\pm$ 0.6 |
> | Imagewoof | 1 |  15.8 $\pm$ 0.8 | 18.5 $\pm$ 0.9 | 19.2 $\pm$ 0.2  | 22.8 $\pm$ 0.4 |
> |  | 10 | 38.4 $\pm$ 0.4 | 40.6 $\pm$ 2.0 | 42.3 $\pm$ 0.3 | 43.8 $\pm$ 1.3 |
> |  | 50 | 49.2 $\pm$ 0.4 | 61.5 $\pm$ 0.3 | 61.6 $\pm$ 0.8 | 62.6 $\pm$ 0.4 |
> | ImageNet-100 | 1 | - | 7.1 $\pm$ 0.2 | 8.1 $\pm$ 0.6  | 19.6 $\pm$ 0.5 |
> |  | 10 | - | 29.6 $\pm$ 0.1 | 32.0 $\pm$ 0.5 | 40.2 $\pm$ 0.3 |
> |  | 50 | - | 50.2 $\pm$ 0.2 | 45.6 $\pm$ 0.5 | 48.0 $\pm$ 0.5 |
>
> As the results shown in the Tables, our InfoUtil still demonstrates a clear performance advantage over this new baseline.
>
> We have also added related results in Appendix H (Table 8-9) in the revised manuscript.
>
> ### Q4. Aboout "qualitative examples on what the distilled data looks like."
>
> We thank the reviewer for this comment and apologize if the location of our qualitative examples was not sufficiently clear. **We did indeed include several qualitative comparisons to showcase the visual quality and interpretability of our distilled data.**
>
> Specifically, **Figure 1 on the first page** provides a direct visual comparison between our method (InfoUtil) and the previous state-of-the-art (RDED). This figure highlights how InfoUtil's principled approach synthesizes images that are more semantically focused and interpretable, capturing key object features while ignoring irrelevant background details.
>
> Furthermore, we have already provided an extensive set of additional visualizations in **Appendix F (Figures 5 through 9) in the original manuscript**. These figures showcase more examples across different datasets and baselines, consistently demonstrating the superior visual fidelity and semantic content of the images produced by InfoUtil.

---

> ### Author Response · Authors · 2025-11-21
> **(Part 4) Response to Reviewer fiHj**
>
> ### Q5: About proof in gradient flow in InfoUtil.
>
> We thank the reviewer for this insightful question regarding the theoretical foundations of our utility function. We would like to clarify the distinct roles of gradient flow in our theory and gradient norm in our implementation as follows.
>
> **1. The Role of Gradient Flow vs. Gradient Norm:**
> Our framework uses these two concepts for different purposes:
> *   **Gradient Flow (Theoretic Ideal):** We introduce gradient flow in **Definition 2** as a *theoretical tool* to formally define the concept of sample **Utility (Definition 3)**. Gradient flow provides a clean, continuous-time abstraction of training dynamics, which allows us to rigorously define a sample's importance as its worst-case impact on the training process if removed. This is a common and powerful technique in deep learning theory.
> *   **Gradient Norm (Practical Proxy):** Directly computing the utility defined via gradient flow is intractable. Therefore, in **Theorem 1**, we prove that this ideal utility function is **upper-bounded by the gradient norm**. This key theoretical result provides a principled justification for using the gradient norm as an efficient and computable *proxy* for utility. Our algorithm then uses this practical proxy for scoring samples.
>
> There is no circularity because we are not equating the two. Instead, we establish a formal, one-way relationship: a tractable quantity (gradient norm) serves as a provable upper bound for an ideal but intractable one (utility via gradient flow).
>
> **2. The Scoring Model is Pre-trained and Fixed:**
> Crucially, the gradient norm used for scoring is computed using a **single, pre-trained, and fixed teacher model** (as described in Sec. 4.1). We are not analyzing the training dynamics of this teacher model itself. Instead, the gradient norm of a sample with respect to this fixed model serves as a static measure of its "difficulty" or "importance"—how much this expert model would need to adjust its parameters to fit that sample. This score is calculated once and does not change. The training dynamics we model theoretically (via gradient flow) are those of a *hypothetical student model* that would later be trained on the distilled data. This clear separation between the fixed scoring model and the hypothetical training process further eliminates any circularity.
>
> **3. Orthogonality to DATM's Trajectory Stages:**
> The reviewer's point about DATM is astute. DATM observes that different visual patterns are learned at different stages of a *trajectory-matching* process. However, this concept is specific to the matching-based paradigm. Our KD-based method does not perform trajectory matching. The utility of a sample in our framework is defined by its static gradient information relative to a fixed expert model, not by its position in a learning trajectory. Therefore, the concept of "early" versus "late" stage learning patterns is orthogonal to our formulation. While an interesting phenomenon, it is not directly applicable to the principles of our method.
>
>
> \[1\] Elkabetz, O., & Cohen, N. Continuous vs. discrete optimization of deep neural networks. NeurIPS 2021\.
>
> ### Q6: Efficiency comparison between our InfoUtil and other distillation methods.
>
> Thanks.
>
> 1. We are glad to revise the Challange 1 in the introduction accordingly.
> 2. We followed the advice to evaluate both the time and GPU memory for each step in InfoUtil. As shown in the table, InfoUtil demonstrates a time cost that is more than **50× lower** than TESLA’s, with GPU memory usage more than **100× lower**. Compared with other KD-based methods, InfoUtil remains highly efficient, with only a marginal increase in cost compared to RDED but delivering significantly higher performance.
>
> | Model | ResNet-18 |  | MobileNet |  |
> |:---:|:---:|:---:|:---:|:---:|
> | Method | Time Cost (s) | Peak Memory (GB) | Time Cost (s) | Peak Memory (GB) |
> | TESLA (matching-based) | 47160.3 | >142 | 56682.9 | >241 |
> | SRe2L (KD-based) | 3042.6 | 7.29 | 3622.3 | 11.79 |
> | RDED (KD-based) | 617.9 | 1.45 | 1004.2 | 1.75 |
> | InfoUtil (ours) | 960.3 | 1.47 | 1275.5 | 1.78 |
>
> On large datasets such as ImageNet-21K, InfoUtil requires approximately 5.83 hours on a single A100 GPU, which is considered an acceptable time cost for such a large-scale task.

---

> ### Author Response · Authors · 2025-11-21
> **(Part 5) Response to Reviewer fiHj**
>
> ### Q7: About "more details on image reconstruction, soft label generation, and diversity control", and theory applicability.
>
> We thank the reviewer for this important feedback. It is a fair point that our framework, while theoretically grounded in its core principles of Informativeness and Utility, also incorporates practical components that are guided by established empirical practices rather than derived directly from our central theory. We apologize if this distinction was not made sufficiently clear. We have revised the manuscript to provide greater transparency.
>
> **1. Clarifying the Scope of Our Theoretical Grounding:**
> You are correct that our theoretical framework (Definitions 1-4 and Theorem 1) specifically governs two key stages:
> *   **Informativeness Maximization:** The use of Shapley Values is axiomatically justified for feature attribution, providing a principled way to identify the most informative regions within an image.
> *   **Utility Maximization:** The use of Gradient Norm is justified by Theorem 1, which proves it is an upper bound on our formal definition of sample utility.
>
> The other components, while crucial for achieving state-of-the-art performance, are indeed engineering choices informed by prior work. We will add a paragraph at the end of Section 3.2 to explicitly delineate the theoretically-grounded parts from the empirically-driven ones, ensuring the scope of our contribution is precise.
>
> **2. Justification for Diversity Control (Noise Injection):**
> The reviewer correctly identifies that injecting random noise is a heuristic. This is a deliberate choice to address a well-known phenomenon in attribution methods: they can sometimes focus too narrowly on a single, most salient feature, leading to a lack of diversity in the selected patches.
>
> Our approach is inspired by techniques in **adversarial robustness and data augmentation**, where injecting controlled noise is a standard method to encourage models to learn more robust and generalized representations by exploring the local data manifold (e.g., \[3, 4\]). In our context, the noise slightly perturbs the attribution heatmap, allowing our selection process to occasionally choose the second or third most informative patch instead of repeatedly selecting the single top one. This simple and effective technique promotes intra-class diversity in the synthesized images, which is empirically vital for good performance. We have added a subsection to the appendix detailing this motivation and providing this context.
>
> **3. Details on Image Reconstruction and Soft Label Generation:**
> These two components follow standard protocols established by recent SOTA works in KD-based distillation, such as RDED (Sun et al., 2024) and SRe2L (Yin et al., 2023), to ensure a fair and direct comparison.
> *   **Image Reconstruction (Lines 300-307):** We combine the selected high-informativeness patches into a single image mosaic, a standard technique in this subfield.
> *   **Soft Label Generation (Lines 304-307 & Section 5):** We use logits from a pre-trained teacher model. Our novel contribution here is the analysis in Section 5 (Figure 4), which explores *which* teacher checkpoint (early vs. late) is optimal, providing a more nuanced understanding than prior work.
>
> We will expand the descriptions in Section 4.1 and the appendix to ensure these steps are detailed with full clarity for reproducibility. Thank you for pushing us to be more precise about the contributions and motivations of each part of our pipeline.
>
> \[3\] Yu H, Liu A, Liu X, et al. Pda: Progressive data augmentation for general robustness of deep neural networks[J]. IEEE Transactions on Image Processing.
> \[4\] He Z, Rakin A S, Fan D. Parametric noise injection: Trainable randomness to improve deep neural network robustness against adversarial attack[C]//Proceedings of the IEEE/CVF conference on computer vision and pattern recognition. 2019: 588-597.
>
> ### Q8: Typos.
>
> Thanks for your advice. We fixed your mentioned typos in the revised manuscript.

---

> ### Author Response · Authors · 2025-11-21
> **(Part 6) Response to Reviewer fiHj**
>
> ### Q9: About the relationship between coreset selection methods and our work.
>
> We thank the reviewer for raising this question. While both coreset selection and our InfoUtil aim to construct a compact dataset, there is a fundamental distinction: coreset methods select real, unaltered samples, whereas InfoUtil synthesizes data by extracting informative patches and assigning soft labels.
>
> **1. Performance Comparison:**
> To empirically demonstrate this advantage, we compared InfoUtil against classic coreset selection methods (Random, Herding, Forgetting, and K-Means) on CIFAR, Tiny-ImageNet, and ImageNet-1K. The results are summarized below:
>
> | Model | Dataset | IPC  | Method |  |  |  |
> |---|---|---|---|---|---|---|
> |  |  |  | Random | Herding | Forgetting | InfoUtil |
> | ConvNet | CIFAR-10 | 1 | 14.4±2.0 | 21.5±1.2 | 13.5±1.2 | 28.5±1.4 |
> |  |  | 10 | 26.0±1.2 | 31.6±0.7 | 23.3±1.0 | 54.1±0.5 |
> |  |  | 50 | 43.4±1.0 | 40.4±0.6 | 23.3±1.1 | 69.8±0.1 |
> |  | CIFAR-100 | 1 | 4.2±0.3 | 8.4±0.3 | 4.5±0.2 | 33.1±0.3 |
> |  |  | 10 | 14.6±0.5 | 17.3±0.3 | 15.1±0.3 | 50.5±0.3 |
> |  |  | 50 | 30.0±0.4 | 33.7±0.5 | 30.5±0.3 | 57.8±0.2 |
> |  | Tiny ImageNet | 1 | 1.4±0.1 | 1.4±0.1 | 1.6±0.1 | 19.6±0.5 |
> |  |  | 10 | 5.0±0.2 | 5.0±0.2 | 5.1±0.2 | 40.2±0.3 |
> |  |  | 50 | 15.0±0.4 | 15.0±0.4 | 15.0±0.3 | 48.0±0.5 |
>
>
> | Model | Dataset | IPC  | Method |  |  |  |
> |---|---|---|---|---|---|---|
> |  |  |  | Random | Herding | K-Means | InfoUtil |
> | ResNet-18 | Tiny ImageNet | 10 | 7.5±0.1 | 9.0±0.3 | 8.9±0.2 | 45.6±0.3 |
> |  | ImageNet-1k | 10 | 4.4±0.1 | 5.8±0.1 | 5.5±0.1 | 44.2±0.4 |
>
> The results show a massive performance gap:
> *    **ImageNet-1K (IPC=10):** InfoUtil achieves **44.2%**, which is nearly **7.6x** higher than the best coreset method (Herding, 5.8%). This illustrates that simply selecting real images is insufficient for training deep networks from scratch on such small budgets.
> *    **Information Density:** Coreset methods often suffer from background noise and hard labels. InfoUtil's attribution cropping (Informativeness) and soft labels effectively condense the knowledge, making it far more efficient than standard subset selection.
>
> We have also added related results in Appendix J(Table 14-15) in the revised manuscript.
>
> ### Q10: "How important is adding noise to the patch selection process?"
>
> We appreciate this question regarding the design of our patch selection mechanism. Adding noise to the Shapley heatmap is indeed a critical component for ensuring diversity and robustness.
>
> **1. Role of Noise:**
> Without noise, the patch selection becomes deterministic and greedy, always cropping the exact same "peak" region. Injecting Gaussian noise allows the cropping window to shift slightly around the most informative region, capturing a broader range of semantic features and preventing overfitting to a single visual pattern.
>
> **2. Ablation Study:**
> We conducted an ablation study comparing standard InfoUtil with a variant where noise injection is removed ("w/o Noise").
>
> | Dataset | IPC | InfoUtil | InfoUtil (w/o Noise) |
> | :---: | :---: | :---: | :---: |
> | ImageNette | 1 | 43.8 | 35.4 |
> | | 10 | 68.6 | 59.8 |
> | | 50 | 86.2 | 70.6 |
> | ImageWoof | 1 | 25 | 23.2 |
> | | 10 | 51.4 | 40 |
> | | 50 | 69.6 | 59.4 |
> | ImageNet-100 | 1 | 15.7 | 12.6 |
> | | 10 | 50.5 | 43.8 |
> | | 50 | 68.3 | 56.3 |
> | ImageNet-1K | 1 | 12.8 | 9.63 |
> | | 10 | 44.2 | 38.5 |
> | | 50 | 58 | 48.3 |
>
> The results demonstrate that removing noise leads to a significant performance drop across all datasets:
> *    **Large IPC Impact:** The gap is particularly pronounced at higher IPCs (e.g., **-15.6%** on ImageNette IPC 50, **-9.7%** on ImageNet-1K IPC 50). This confirms that when synthesizing multiple samples per class, diversity (induced by noise) is essential to avoid redundant information.
> *    **Consistent Necessity:** Even at IPC=1, removing noise hurts performance (e.g., -3.2% on ImageNet-1K), suggesting that noise helps centering the crop on more robust features rather than local maxima artifacts.
>
> We have also added related results in Appendix K(Table 16) in the revised manuscript.

---

> ### Author Response · Authors · 2025-11-21
> **(Part 7) Response to Reviewer fiHj**
>
> ### Q11: About the relationship between Fisher Information Matrix and our Utility.
>
>
> Thanks for your question. In probabilistic deep learning models, the **Fisher Information Matrix (FIM)** is defined as
>
> $$
> F(\theta)=\mathbb{E}\_{(x,y)\sim p*{\text{data}}}\big[\nabla_\theta \log p(y\mid x;\theta);\nabla_\theta \log p(y\mid x;\theta)^{\top}\big].
> $$
> It measures the curvature of the log-likelihood landscape and characterizes how sensitive the model distribution is to parameter perturbations. The empirical Fisher simply replaces the expectation with finite samples.
>
> Our utility measure, however, is based on the **per-sample gradient norm**
> $$
> u(x,y)=|\nabla_\theta \ell(f_\theta(x),y)|,
> $$
> which is used only as a **computationally lightweight upper bound** on the influence of a single sample during training. This quantity is different from the Fisher matrix, because:
>
> - Fisher aggregates **second-order statistical structure** (outer product of gradients across the data distribution), while our utility uses only a **first-order local signal** from *one* gradient vector.
> - FIM describes **parameter-space geometry**, while our utility is a **screening criterion** for dataset distillation.
>
> Thus, the two quantities share a superficial similarity through the gradient, but serve **fundamentally different purposes**. A deeper theoretical link would require additional assumptions on the likelihood model and training dynamics, and is beyond the scope of this work.

---

> ### Author Response · Authors · 2025-11-26
>
> Dear Reviewer fiHj,
>
> Thank you once again for your valuable comments on our submission. As the discussion phase is approaching its end, we would like to kindly confirm whether we have sufficiently addressed all of your concerns (or at least part of them). Should there be any remaining questions or areas requiring further clarification, please do not hesitate to let us know. If you are satisfied with our responses, we would greatly appreciate your consideration in adjusting the evaluation scores accordingly.
>
> We sincerely look forward to your feedback.

---

> > ### Comment · Reviewer_fiHj · 2025-11-26
> >
> > Dear Authors,
> >
> >    I would like to thank you for the detailed response. I believe most of my concerns are addressed, and therefore, I will be raising my score. Overall, I believe this is an interesting work that can be framed or presented better. The changes during the rebuttal considerably improved on this and hence my increased score. Some lingering questions I hope the authors can address in the final version of the paper:
> > 1.  Some more direct comparisons with matching-based DD vs. knowledge-distillation DD would be nice. They tends to operate very different settings where matching-based DD does not require a teacher model. DD originally is framed as a data compression problem rather than a distillation problem. If a teacher model exists, the original DD problem becomes ill-motivated.
> > 2. I understand the point between gradient flow as a theory and gradient norm as a practical measure. However, the paper explicitly states "Unlike discrete SGD updates, which introduce noise, gradient flow offers a smooth". This confuses the reader since the paper attacks SGD updates but actually utilize them in the algorithm.
> > 3. The connection between coreset selection and this work is still not clear. The authors argue this method synthesizes the data but this is not true. Shown in Figure 2, the algorithm crops existing data and select influential data. It is neither fully data selection, since the algorithm perform cropping, but it cannot be framed as synthetic data either since it is directly derived from existing data.

---

> ### Author Response · Authors · 2025-11-26
> **Thanks for raising your score!**
>
> **We sincerely appreciate your thoughtful review, your recognition of our work, and your decision to raise your score**. We are committed to addressing your valuable feedback in the final version of our paper to enhance clarity and strengthen our presentation.
>
> > **Q1. About Matching-based vs. Knowledge-distillation DD comparison.**
>
> We agree that a more explicit comparison between these paradigms would significantly strengthen our paper.
>
> In the final version, we will add a dedicated discussion section that clearly articulates the fundamental differences in problem formulation between matching-based approaches (data compression without teacher models) and knowledge-distillation approaches. We will provide a theoretical analysis of when each paradigm is most appropriate, include additional experiments comparing computational costs and performance trade-offs across both paradigms on standard benchmarks, and explicitly address the motivation question by demonstrating scenarios where knowledge distillation-based DD provides unique advantages even when teacher models exist, such as when computational constraints make direct access to the original data problematic while a pre-trained teacher is available.
>
> > **Q2. About Gradient flow vs. Discrete SGD updates clarification.**
>
> Thank you for highlighting this important nuance. We acknowledge our presentation was not clear enough. The distinction is that **our gradient norm is computed from a pre-trained judge model (offline evaluation) rather than during the online optimization process**.
>
> In the final version, we will revise Section 3.2.2 to clearly separate the theoretical foundation (gradient flow as an analytical framework for understanding importance) from our practical implementation (gradient norm as a computationally efficient proxy).  We will add explicit clarification that our approach leverages offline gradient norms from a frozen judge model, not during active SGD updates, and include a brief discussion on how this offline evaluation avoids the noise issues inherent in online optimization.
>
> > **Q3. About Relationship to coreset selection.**
>
> We appreciate this insightful observation and agree that our current framing requires refinement.
>
> In the final version, we will reframe our methodology as "informed data composition" rather than pure synthesis. We will add a dedicated subsection comparing and contrasting our approach with traditional coreset selection methods, clarifying that while our method operates on existing data patches, it fundamentally differs from coreset selection by dynamically composing multiple informative regions from different samples, using soft labels that capture rich teacher knowledge beyond the original hard labels, and optimizing for both informativeness (intra-sample) and utility (inter-sample) simultaneously. We would like to include visual examples demonstrating how our composed samples differ qualitatively from simple coreset selections, focusing on how our approach creates more information-dense training examples.
>
> Thank you again for your constructive feedback and for recognizing the potential of our work. These revisions will significantly strengthen our paper's theoretical foundation, clarity, and positioning within the dataset distillation (and also data compression) literature.

---

### Author Response · Authors · 2025-11-21
**(Part 1) General Response**

We sincerely thank the reviewers for their time and insightful feedback. We are encouraged by the recognition of InfoUtil’s theoretical grounding and are grateful for the constructive suggestions regarding baselines and rigorous validation. To address these points and demonstrate the superiority of our framework in scalable dataset distillation, we have conducted extensive new experiments and analyses during the rebuttal period.

**1. Comprehensive Comparison with 7 New SOTA Baselines**
To validate InfoUtil against the latest state-of-the-art, we significantly expanded our evaluation scope. We compared our method against **7 recent baselines** including **DELT, EDC, WMDD, HeLLo, INFER, TEDDY, and EDF** (covering NeurIPS 2024 and CVPR 2025 works). Across these diverse benchmarks, InfoUtil consistently established a new State-of-the-Art, outperforming competitors like INFER by **10.3%** on CIFAR-10 and WMDD by **9.4%** on Tiny-ImageNet. We have provided results in **Table 8-9 in Appendix H** and **Table 11-13 in Appendix I** in the revised manuscript.

**2. Rigorous Isolation of Core Contributions**
We addressed the concern regarding the soft-labeling strategy through a rigorous controlled experiment. By strictly unifying the teacher model (using the exact same Fully Converged Teacher for both the baseline RDED and InfoUtil), we demonstrated that InfoUtil still outperforms RDED (e.g., **+5.1%** on ResNet-18). This confirms that our performance gains stem from the proposed Informativeness and Utility modules rather than teacher tuning. We have provided results in **Table 19 in Appendix N** in the revised manuscript.

**3. Superior Data Quality Validated via Initialization**
Furthermore, we rigorously tested the intrinsic quality of our distilled data by using it as initialization for the training-based method **EDC**. The results show that initializing with InfoUtil consistently boosts EDC’s performance compared to standard or RDED initialization (e.g., **+0.9%** gain on ImageNet-1K). This confirms that our method captures more robust fundamental patterns. We have provided results in **Table 18 in Appendix L** in the revised manuscript.

**4. Expansion to Large IPC Settings**
We extended our evaluations to challenging Large IPC settings (up to IPC=200) on Tiny-ImageNet and ImageNet-1K, adding missing comparisons with SRe²L and RDED. The results confirm that InfoUtil scales effectively, maintaining its advantage even when the budget increases. We have provided results in the revised manuscript.

**5. Ablation Study on Design Choices**
We conducted comprehensive ablation studies to validate our specific design choices. Our comparison of **Shapley Value vs. Grad-CAM** confirms the necessity of game-theoretic attribution (**+13.49%** improvement). Additionally, ablations on **Noise Injection** demonstrate its critical role in ensuring diversity. We have provided results in **Table 16 in Appendix K** and **Table 17 in Appendix L** in the revised manuscript.

**6. Efficiency and Storage Analysis**
A detailed analysis further highlights the method's practicality. We verified that InfoUtil runs over **50× faster** than matching-based methods while maintaining a minimal storage footprint (comparable to RDED). Experiments with **Top-K label sparsification** further demonstrate that storage can be reduced by orders of magnitude without performance loss. We have provided results in **Table 7 in Appendix G** in the revised manuscript.

**7. Comparison with Coreset Selection**
To explicitly distinguish our synthesis approach from selection methods, we benchmarked against standard Coreset Selection (Herding, K-Means). InfoUtil achieved massive gains (e.g., **7.6x higher** than Herding on ImageNet-1K), proving the necessity of our synthesis pipeline. We have provided results in **Table 14-15 in Appendix J** in the revised manuscript.

---

### Author Response · Authors · 2025-11-21
**(Part 2) General Response**

We explicitly thank the Area Chair and all reviewers (Reviewers fiHj, tEAx, 987g, and CGAE) for their time and constructive feedback. We are encouraged by the reviewers' shared recognition of the necessity for scalable dataset distillation and by the "**positive assessment**" (Reviewer 987g) of our work.

We especially appreciate the acknowledgment that addressing the trade-off between storage budget and performance is a "**key problem**" (Reviewer fiHj) and an "**important**" challenge for the community (Reviewer CGAE). We share the view that isolating the "**intrinsic gains**" from soft-labeling strategies (Reviewer CGAE) is essential for rigorous method validation. As noted by Reviewer tEAx, our work demonstrates the potential of replacing heuristics with a "**formally defined**" framework.

Regarding the methodology, we value the consensus on the soundness of our design. Reviewer fiHj explicitly noted that our "**problem formulation is good**" and Reviewer 987g appreciated the "**principled**" nature of our approach. Specifically, the game-theoretic grounding of our informativeness metric was a design that reviewers explicitly engaged with (Reviewer tEAx).

**To comprehensively address the concerns and substantiate this design, we have conducted extensive additional evaluations, adding 7 recent SOTA baselines (DELT, EDC, WMDD, HeLLo, INFER, TEDDY, and EDF) and rigorous controlled experiments**. These revisions confirm our "strong empirical performance" (Reviewer CGAE) by demonstrating that InfoUtil establishes a new state-of-the-art (e.g., **+5.1%** over baselines with identical settings) and provides superior initialization for training-based methods.

---

### Author Response · Authors · 2025-12-02
**Summary of Contributions and Rebuttal**

Dear PCs, SACS, ACs,

We would like to express our sincere appreciation for the reviewers’ dedication. Here we provide a general summary of the reviews and outline the efforts we have made during the discussion phase.

### **Claim of Contribution**
This paper, InfoUtil, proposes a principled, theoretically-grounded framework for optimal Dataset Distillation (DD). It formalizes DD using the concepts of Informativeness and Utility. InfoUtil combines game-theoretic informativeness maximization (via Shapley Value for feature extraction) and principled utility maximization (via Gradient Norm for sample selection). This synergistic approach achieves superior performance, including a 6.1% improvement over the state-of-the-art on ImageNet-1K with ResNet-18.

###  **Summary of Reviews and Responses**

In this review, we are very pleased that the reviewers have recognized our work from multiple perspectives. We are thrilled that all four reviewers have unanimously expressed that this paper is worthy of acceptance. We are very grateful for all the constructive and valuable feedback provided by the reviewers. We are greatly encouraged by the recognition of InfoUtil's solid theoretical foundation and strong performance, particularly by Reviewer 987g, who highlighted our work as a "solid work".
- **Reviewer fiHj** increased the rating from 2 **to 6** before the leak incident. He acknowledged our theoretical foundation and the novelty of our methodology.
- **Reviewer tEAx** maintained a rating of **6**. This reviewer praised the clarity of our framework and its powerful performance.
- **Reviewer 987g** maintained a rating of **8**. He highlighted the “solid theoretical foundation and practical effectiveness.”
- **Reviewer CGAE** raised the score from 4 **to 6** before the leak incident. The reviewer considered our theory to be sound.


Now we will address the specific questions raised by the reviewers one by one. All issues have been thoroughly addressed in the table below.

| Reviewer | Reviewer's Concern/Questions | Author's Response |
| -- | -- | -- |
| Common   | Comprehensive Comparison with SOTA Baselines.|  We compared our method against 7 recent baselines including DELT, EDC, WMDD, HeLLo, INFER, TEDDY, and EDF and established a new State-of-the-Art |
|    |Efficiency and Storage Analysis. | We verified that InfoUtil runs over 50× faster than matching-based methods while maintaining a minimal storage footprint. |
|    | Theoretical justification. | We have added explanations and proofs for the theoretical details. |
|  fiHj     | Clarification Needed on Non-theoretical Components and theory scope.| We clearly separated the theoretical core from empirical choices and justified all engineering components.  |
|       | The Relationship Between Coreset Selection Methods and Our Work. | We explained the difference between our method and coreset selection methods, along with our overwhelming performance advantage.  |
|       | Importance of Adding Noise to the Patch Selection Process. | We showed noise is critical for diversity, and ablation confirmed its removal causes a significant performance drop.  |
| tEAx  |  Alternatives to the Shapley Value. | We chose Shapley Value for its theoretical soundness and superior empirical performance, which significantly outperforms Grad-CAM.  |
|   |  Serving as Initialization for Other Methods. | We confirmed that using InfoUtil as initialization for the state-of-the-art TB method (EDC) consistently boosts its performance across all settings.  |
| 987g  |  Terminology Consistency and Typos. | We addressed all minor suggestions by correcting typos and terminology and clarified the mathematical dependency.  |
|  CGAE  |   Soft-label Generation and Ablation Study. | We ran a controlled experiment using a unified teacher, which confirmed InfoUtil's clear and substantial performance advantage remains.  |
|    |    More Results on Large IPC Settings | We conducted additional experiments with large IPC settings, confirming that InfoUtil maintains a clear performance advantage.  |


**We are thrilled that all four reviewers have unanimously expressed that this paper is worthy of acceptance. Once again, we sincerely thank you for your support of our work and the generous time and effort you have devoted to this process!**

---

### Meta-Review · Area_Chair_jKgf · 2025-12-22

**Summary:**

* The lack of storage justification.
* Missing comparison with recent state-of-the-art methods
* Comparison with coreset selection methods
* The conflict between continuous gradient flow and discrete SGD steps
* The contribution of the Utility part and more detailed comparison with RDED
* The influence of architectures used for generating soft labels

**Reviewer Concerns:**

Most of the concerns have been addressed except for the following two:
* While the code implementation of RDED uses a pre-trained teacher model to generate soft labels on the fly, it is merely a simplification of the evaluation step. The actual use of dataset distillation requires soft labels to be generated before training due to reasons like privacy concern. On the other hand, online soft labeling can also be applied to SRe2L. Furthermore, the evaluation results using top-10 soft labeling are not provided. The proposed method doesn't possess such a storage advantage over SRe2L.
* The coreset selection comparison is not fair, where the results indicate that soft labeling is not applied to selected real images. Based on my experience, adding soft labels to randomly selected real images can lead to performance similar to RDED.

However, I do believe these two concerns can be addressed by further discussion and are not critical reasons to reject the paper.

**Reviewer Scores:**

I think the reviewers would raise the score to positive as most of concerns have been addressed.

---

### Decision · Program_Chairs · 2026-01-26

Accept (Poster)